# Drone-based GPR application to snow hydrology

Eole Valence[1,2], Michel Baraer[1,2], Eric Rosa[2,3], Florent Barbecot[2,4], Chloe Monty[1]

[1]Hydrology, Climate and Climate Change (HC[3]) Laboratory, Ecole de Technologie Superieure, Montreal, H3C 1K3, Canada
[2]Geotop, Montreal, H2X 3Y7, Canada
[3]Groupe de recherche sur l'eau souterraine, Universite du Quebec en Abitibi-Temiscamingue, Rouyn-Noranda, J9X 5E4, Canada
[4]Departement des sciences de la Terre et de l'atmosphere, Universite du Quebec a Montreal, Montreal, H2L 2C4, Canada

*Correspondence to*: Eole Valence (eole.valence.1@ens.etsmtl.ca)

**Abstract.** Seasonal snowpack deeply influences the distribution of meltwater among watercourses and groundwater. During rain-on-snow (ROS) events, the structure and properties of the different snow and ice layers dictate the quantity and timing of water flowing out of the snowpack, increasing the risk of flooding and ice jams. With ongoing climate change, a better understanding of the processes and internal properties influencing snowpack outflows is needed to predict the hydrological consequences of winter melting episodes and increases in the frequency of ROS events. This study develops a multi-method approach to monitor the key snowpack properties in a non-mountainous environment in a repeated and non-destructive way. Snowpack evolution during the winter of 2020–2021 was evaluated using a drone-based, ground-penetrating radar (GPR) coupled with photogrammetry surveys conducted at the Sainte-Marthe experimental watershed in Quebec, Canada. Drone-based surveys were performed over a 200 m² area with a flat and a sloped section. In addition, time domain reflectometry (TDR) measurements were used to follow water flow through the snowpack and identify drivers of the changes in snowpack conditions, as observed in the drone-based surveys.

The experimental watershed is equipped with state-of-the-art automatic weather stations that, together with weekly snow pit measurements over the ablation period, served as a reference for the multi-method monitoring approach. Drone surveys conducted on a weekly basis were used to generate georeferenced snow depth, density, snow water equivalent and bulk liquid water content maps.

Despite some limitations, the results show that the combination of drone based GPR, photogrammetric surveys and TDR is very promising for assessing the spatiotemporal evolution of the key hydrological characteristics of the snowpack. For instance, the tested method allowed for measuring marked differences in snow pack behaviour between the first and second weeks of the ablation period. A ROS event that occurred during the first week did not generate significant changes in snow pack density, liquid water content and water equivalent, while another one that happened in the second week of ablation generated changes in all three variables. After the second week of ablation, differences in density, LWC and SWE between the flat and the sloped sections of the study area were detected by the drone-based GPR measurements. Comparison between different events was made possible by the contact-free nature of the drone-based measurements.

## 1 Introduction

By acting as transient storage, seasonal snow cover determines the amplitude of spring floods, the level of late summer flows and the recharge of aquifers (Dewalle and Rango, 2008). Snowmelt floods are a cause of economic losses and sometimes loss of life (Ding et al., 2021), while insufficient aquifer recharge affects water availability for agricultural and industrial uses, fresh water supply and the ecology of river systems (Dierauer et al., 2021).

Recent changes in snow cover characteristics have been reported from different regions of the globe (Magnusson et al., 2010; Zhang et al., 2015), (Hodgkins and Dudley, 2006; Cho et al., 2021; Ford et al., 2021; Najafi et al., 2017). Climate change projections anticipate further alteration of snowpack characteristics: seasonal snowpack depth is expected to diminish (Dierauer et al., 2021), the winter maximum snow water equivalent to decline (Sun et al., 2019) and the spring melt to occur

earlier in the season (Gergely et al., 2010). Moreover, observations and models indicate an increase in the number of winter rain-on-snow (ROS) events (Li et al., 2019). Combined with changes in snowpack characteristics, those events are predicted to trigger increases in winter flood and ice jam intensity and frequency (Morse and Turcotte, 2018; Andradóttir et al., 2021).

Within this context, monitoring the spatiotemporal evolution of snow cover properties appears essential for anticipating adverse climate change consequences on winter hydrology and groundwater recharge (Lindström et al., 2010).

Snow depth ($h$), snow water equivalent ($SWE$), density ($\rho$), and liquid water content ($LWC$) are among the most measured properties of the snowpack (Kinar and Pomeroy, 2015). These four variables are considered key properties for characterizing the snowpack's hydrological behaviour (Vionnet et al., 2021). Different technics have been developed over time to

independently monitor those four variables over very limited surfaces (less than 100 m$^2$ for most of them):

- Snow depth is widely monitored using ultrasonic sensors (Doesken et al., 2008), and methods like global navigation satellite system interferometric reflectometry (GNSS-IR) (Chen et al., 2021) and terrestrial laser scanning (Prokop, 2008; Revuelto et al., 2015; Deems et al., 2017) are gaining in popularity. Still, destructive manual measurements remain extensively used for snow depth surveying (Leppänen et al., 2016).

- $SWE$ can be calculated based on manual snow-coring to estimate sample volume and mass. The manual method is time-consuming, destructive and of moderate precision (Goodison et al., 1987; Morris and Cooper, 2003; Sturm and Holmgren, 2018; Paquotte and Baraer, 2022). Automatic monitoring makes it possible to capture $SWE$ temporal variability. The methods most often used are gamma ray monitoring (GMON), cosmic ray neutron probe (CNRP), snow pillows and plates, the system for acoustic sensing of snow (SAS2), the snowpack analyzer (SPA-2) and GNSS

receiver-based SWE estimators (Yu et al., 2020). Most of those technics require site calibration.

- Snow density is commonly measured through gravimetric measurements or calculated from snow depth and $SWE$ measurements (Conger and Mcclung, 2009). In dry conditions, snow density can be estimated with a dielectric permittivity measurement system such as the Finnish SnowFork (Hao et al., 2021). Other methods include neutron probes (Hawley et al., 2008) and diffuse near-infrared transmission (Gergely et al., 2010).

- The most common in situ $LWC$ measurement methods are based on snow permittivity measurements. The SnowFork (Sihvola and Tiuri, 1986), the Denoth device (Denoth, 1995) and the A2 Photonic WISe sensor (Webb et al., 2021) are among the most popular devices to measure $LWC$. They all have an accuracy level of around 1% of the volumetric $LWC$. The most accurate method, however, which is often used as a reference for those devices, is freezing or melting calorimetry (Webb et al., 2021; Mavrovic et al., 2020). $LWC$ may be monitored unattended using time domain

reflectometry (TDR), but multiday monitoring using that technique still presents a challenge (Lundberg et al., 2016).

Even if they are accurate in following the evolution of each variable in time, those techniques do not allow for capturing the spatial variability in the snowpack properties unless they are repeated at a multitude of points. Aerial and spaceborne remote sensing represents an attractive alternative for that purpose.

With a vertical accuracy of less than 10 centimetres, airborne photogrammetry allows for a non-destructive monitoring of the spatial variability of snow depth in open areas (Bühler et al., 2016a; Avanzi et al., 2018; Harder et al., 2020; Jacobs et al., 2021). In forested areas, airborne lidar (light detection and ranging) has proven a more accurate option (Koutantou et al., 2021; Dharmadasa et al., 2022). The use of satellite-based remote sensing for *snow depth* and, by extension, *SWE* mapping has received much attention over the past decade (Guneriussen et al., 2001; Rott et al., 2003). While showing promising results

and fast improvements in large open areas of several square kilometres range (e.g. Mcgrath et al. (2019)), satellite-based *SWE* and/or snow depth estimations still involves coarse spatial data with a high degree of uncertainty when passive sensors are used (Mortimer et al., 2020), and some accuracy challenges still exist with active sensors (Pfaffhuber et al., 2017). This is the case in mountainous areas, for example (Liyun Dai, 2022).

From the 1980s onward, the use of ground-penetrating radar (GPR) has been seen as a solution to overcome the difficulties in
capturing key properties of and spatial variability in the snow pack, as described above (Marchand et al., 2003). First carried
by the operator, GPR airborne and ground-vehicle-based applications have risen in popularity due to their abilities to cover
transects that are one tenth of a kilometre long (Bruland and Sand, 1998). Radargrams generated using GPR show the influence
a milieu has on the emitted electromagnetic wave that travels through it. This influence is characterized by the milieu's
permittivity, expressed as a complex number. For snow layers, the real component of the permittivity is mostly a function of
snow density, snow depth and *LWC*. As *SWE* can be calculated from snow depth and density, GPR therefore allows for
measuring a physical characteristic that is related to the four key snowpack properties in a single survey (Di Paolo et al., 2018).
Since the 1980s, GPR has been shown to be a valuable tool for measuring physical snowpack characteristics (Holbrook et al.,
2016). It is one of the most-used methods in snowpack studies (Vergnano et al., 2022), and the spatial variability of snow
properties has been extensively assessed using GPR (Lundberg et al., 2010; Previati et al., 2011; Holbrook et al., 2016).

However, GPR applications in monitoring one or several of the four key snowpack characteristics still involve different
challenges, such as that:

1. The real component of the permittivity requires the snow depth to be known or estimated (Di Paolo et al., 2020).
2. Different empirical equations have been developed to relate snow density and *LWC* to the real component of the
   permittivity (Frolov and Macheret, 1999; Di Paolo et al., 2018). In dry conditions, *LWC* being neglectable, a direct
   relation exists between the snow density and the relative permittivity. On the other hand, the introduction of liquid
   water into the snowpack cannot be accurately characterized with GPR velocity alone (Bradford and Harper, 2006).
   In absence of other measurements allowing for mapping of *LWC* or snow density in wet conditions, either an
   assumption need to be made regarding snow density variability from spot measurements (e.g., Webb et al. (2020);
   Yildiz et al. (2021)) or an empirical relation must be parametrized by calibration (e.g. Singh et al. (2017)).
3. Ground-based GPR applications requires direct contact with the snow surface, modifying its properties (Valence and
   Baraer, 2021) and making subsequent surveys not fully representative of natural conditions.
4.  Air surveys such as the helicopter-based ones are limited by high operating costs, while ground-based surveys are
   difficult to conduct on unstable and steep slopes (Vergnano et al., 2022).

Recent developments show interesting potential to overcome those challenges. Combining GPR applications with other
measurements has been shown to be an efficient way to overcome the two first challenges. For instance, Marchuk and
Grigoryevsky (2021) improved GPR-based snow depth profiling by associating GPR to a laser range finder. The use of drone-
based surface mapping methods such as photogrammetry or lidar in snowpack studies provides reliable snow depth maps
(Bühler et al., 2016b). Lundberg et al. (2016) and Yildiz et al. (2021) used drone-based photogrammetry or lidar to integrate
snow depth measurements into *SWE* calculations. Combining techniques that monitor the temporal evolution of the snow
permittivity, such as TDR, with GPR has been shown to be a promising approach to studying snowpack spatial variability over
a given period (Godio et al., 2018). Estimating *LWC* from frequency-dependent attenuation of the GPR signal, as proposed by
Bradford et al. (2009) is another way to address the wet snowpack characterization issues.

Actual developments in drone-borne GPR have opened new avenues in GPR-based snow pack studies (Francke and
Dobrovolskiy, 2021). Recent studies have shown it to be valuable in snow avalanche applications (Mccormack and Vaa, 2019)
and in snow depth mapping (Tan et al., 2017; Vergnano et al., 2022). Similarly, drone-based ultra-wide-band (Jenssen and
Jacobsen, 2020) and software-defined radar (Prager et al., 2022) applications to snowpack characterization surveys have
recently been demonstrated to be potentially ground-breaking solutions.

The present study aimed to monitor the spatiotemporal variability in snow depth, snow density, *SWE* and snow *LWC* of a snowpack over flat and sloped areas with a non-destructive approach. This objective was achieved by combining some of the emerging solutions described above with more traditional snow-monitoring techniques in a novel way. This combination included drone-based photogrammetry; drone-based GPR; and continuous monitoring of *SWE*, snow depth and snow permittivity using TDR and snow pit-based measurements.

## 2 Study site

The study was conducted at the Bassin versant experimental (BVE) Ste-Marthe, an experimental watershed located approximately 70 km west of Montréal, in Quebec, Canada (45.4239°N, 74.2840°W) (Fig. 1a.). The main station of the BVE Ste-Marthe is situated at 120 m above sea level, in an approximately 200 m$^2$ forest clearing (Fig. 1b. and Fig. 1c.). A distinction is made between two different topographic areas of the clearing: one of approximately 30 m$^2$, categorized as flat, and the other of approximately 50 m$^2$, categorized as sloped (Fig. 1d.). The automatic weather station (AWS) measures various hydroclimatic variables. Those of interest for the purpose of this study are listed in Table 1.

Measurements took place during the winter of 2020-21, from February 26 to March 26. Two rain-on-snow (ROS) events occurred during this period. The first ROS was observed from February 28 to March 1, and the second from March 9 to 12. Field visits for drone-based surveys and snow pit measurements occurred on February 26, March 5, March 12 and March 19. February 26 corresponded to the end of the accumulation period, while the ablation period started February 27.

## 3 Methods

The spatiotemporal variability of snow depth, snow density, *SWE* and snow *LWC* was assessed by combining different methods with different sampling approaches. Table 2 provides a list of the different methods that were used. Those were split into three categories according to the frequency of measurements and the spatial coverage. Repeated surveys conducted over the flat and sloped areas were used to produce maps of the four studied variables on a weekly basis. Continuous and repeated measurements at a single point were used for verification of the map data at a given point in the study area. TDR sensors were an exception: A total of eight probes were split between the two areas. At each spot, probes were placed on different hard layers on the snowpack. These layers were identified as possible vectors for lateral flow (Evans et al., 2016).

### 3.1 AWS monitoring

Data from the AWS were recorded using a Campbell Scientific CR1000 data logger. AWS sensors included the snow lysimeter situated at less than 10 m from the surveyed flat area, and presented a comparable exposition to sunlight.

Therefore, it was expected that both snow depth records would exhibit comparable results. The snow's relative density was calculated using the snow depth and the SWE measurements, following equation 1:

$$SWE = h \times \rho, \tag{1}$$

*SWE* and $h$ are both in metres, and $\rho$ is dimensionless.

The frozen ground depth was estimated by interpolation of ground temperatures measured from 10 to 60 cm below the ground level at 10 cm depth intervals. Snowpack temperature was measured with four thermometers at 0, 10, 20 and 30 cm above the ground level. To avoid using snowpack temperature measurements that could have been influenced by solar radiation or by contact with air, snowpack temperature measurements were not considered after March 17, 2021, the day the snow height decreased below 40 cm above ground level.

## 3.2 Manual measurements

Manual measurements were conducted on a weekly basis, on the same days as the drone surveys. Snow pits were excavated, with northern orientations, at approximately 75 cm from the previous ones, following the method presented by Fierz et al.
(2009). The snow pits were located less than 3 m from the flat area surveyed by drone and presented a comparable exposition to sunlight. For each pit, layer identification was followed by a sequential depth, density and snow temperature measurements. Each layer was isolated from the preceding one using a thin metallic plate and sampled using a metallic cylinder of 0.3 dm$^2$ or a cylindrical plexiglass sampler with a surface of 0.5 dm$^2$. The sample mass was measured in situ with a scale of ± 1 g accuracy. *LWC* measurements were made in snow pit using an A2 Photonic WISe sensor. Two vertical measurement profiles with 10
cm intervals between observations were created for each snow pit. Even if the manufacturer's device precision was ± 1% of *LWC*, we anticipated a higher degree of uncertainty, as measurement through ice layers was not possible.

Snow pack total depth and bulk *SWE* were calculated by adding up the individual layer values for these measures. Bulk relative density and *LWC* were calculated by taking the weighted mean of the measurements of a layer's thickness.

## 3.3 TDR monitoring

The CS610 probes were controlled using a TDR200 (both from Campbell Scientific). Each probe was bench calibrated according to the Campbell Scientific guidelines before deployment. Onsite, each CS610 was inserted into the snow, and left lying over a hard layer without any guide or support. This setup was chosen to allow the probe to move downward together with the supporting hard layer as the snowpack settled. Maintaining probes at a fix position above the ground triggers air pocket formation around the metallic rod over time, affecting the measurements' accuracy (Pérez Díaz et al., 2017). As no
visible differences in stratigraphy were observed between the flat and sloped areas over the accumulation period, hard layers supporting the probes were identified the same way:

- $\alpha$ represents the ground level. Probes were installed January 8 in the flat area and January 26 at the base of the slope. At both locations, the snow layer on top of the ground was unconsolidated and heterogeneous.
- $\beta$ is a wind crust formed December 30–31. Probes were installed on the same days as on layer $\alpha$. The layer $\beta$ was
overlaid by unconsolidated granular snow.
- $\gamma$ is a hard settled snow layer formed on January 15. Probes were installed January 26. At that time, the hard snow layer was overlaid by a thin layer of fresh snow.
- $\varepsilon$ is an ice layer formed after a freezing rain event that occurred February 16. Probes were installed on top of that layer on February 22 at both locations.

TDR probes measure the relative permittivity of the surrounding material. In snow, the relative permittivity is a function of the density and the liquid water content mainly (Stacheder et al., 2009). Placed at different spots, initial relative permittivity values measured by the probes naturally differ slightly from each other. In order to allow comparison of the permittivity evolution between two probes placed over the same layer, relative permittivity values were normalized by dividing all values measured by a TDR probe by the first measurements made after installation of that given probe value. Doing so allowed the
researchers to start each TDR-derived time series from 1. With a 15-minute measurement interval, it was assumed that any sensible variations in permittivity (higher than 10% of the initial value) between two successive measurements was due to changes in liquid water content, as such changes would occur over a longer time scale if due to a change in snowpack density only (Stacheder et al., 2005).

## 3.4 Drone-based photogrammetry

A DJI Mavic 2 Pro drone was used to capture the RGB images used for photogrammetry. During a flight time of approximately 20 minutes, the drone took around 200 images with an 80% overlap at an elevation of 25 m above ground level. The Mavic 2 Pro is equipped with a TopoDrone global navigation satellite system (GNSS) to allow post-processing kinematic (PPK)

treatment to correct images' locations. The uncertainty claimed by the manufacturer is 3–5 cm in all directions. PPK corrections were made using a Reach RS2 GNSS base station, with a manufacturer's uncertainty of 4 mm in the horizontal direction and 8 mm in the vertical. After each site visit, collected images were processed using the Pix4Dmapper software to produce digital surface models (DSMs). The expected horizontal resolution is 0.6 cm per pixel. Vertical accuracy was assessed using ground control points (GCP). For each survey, ten GCP were disposed all around the study area; GCP were placed approximately at the same position for each survey. Control points were geo-localized using a KlauGeomatic 7700B GNSS rover, ensuring a 5-cm accuracy. Comparing the uncorrected DSM models produced by photogrammetry to the DSM models after correction using control points showed variations under 3 cm in all three directions, which is within the expected accuracy of the Klau GNSS. The use of control points did not lead to meaningful improvements of the map georeferencing, validating the accuracy of maps produced using PPK adjustments only.

Finally, snow depth maps were produced by subtracting a snow-free DSM produced on April 6, just after the complete thaw of the snow cover and just before the vegetation growth, from the DSM produced in winter conditions. This was done using the ESRI Geographic Information System (GIS) software ArcGIS, following the protocol presented by Bühler et al. (2016a) and Yildiz et al. (2021).

### 3.5 Drone-based GPR permittivity measurement

GPR surveys were performed using a Radar System Inc. Zond 1.5 GHz carried by a DJI Matrice (M) 600 Pro drone. The GPR integration system and the flight control software (UGCS) were supplied by SPH Engineering. Maximizing GPR measurements requires flying at 1.2 m/s and at approximately 1 m above the surveyed surface. Drone altitude was controlled using a terrain-following system supplied by SPH Engineering. The system is made of the UgCS SkyHub on-board computer coupled with a radar altimeter. The on-board computer manages the power supply and the GPR data. The M600 Pro was equipped with a KlauGeomatic 7700B GNSS allowing position correction via PPK. Similar to the photogrammetry, PPK corrections were made using the Reach RS2 GNSS as a base station. GPR data were referenced by post treatment using the KlauGeomatic PPK solution. Surveys were performed over both the flat and sloped areas. Post-treatment of radargrams was performed with the Radar System Inc. Prism2 software. The GPR system was sampled every 512 ns over both flat and sloped areas, and the drone's flight followed north-south transects. Thus, the spatial resolution of GPR measurements was a function of the actual drone speed (different from expected drone speed due to wind and other meteorological conditions affecting the drone flight) and the sampling frequency. For each survey, six transects on the flat area and nine transects on the slope were surveyed. The distance between two consecutive transects was $50 \pm 20$ cm. Post-treatment consisted of applying a background removal filter, adjusting the gain, and applying a time-delay compensation. The ground/snow and snow/air interfaces were detected automatically wherever possible and manually where the layer boundaries were not recognized by the automatic graphic interpretation tool. Figure 2 provides two examples of radargrams with identified layer boundaries. Given that GPR measurements were geolocalized using the same PPK as the photogrammetry, the radargram georeferencing was considered to have a 5 cm accuracy.

Snow depth for each GPR transect was extracted from the snow depth maps produced by photogrammetry using ArcGIS. The velocity of the electromagnetic wave within the snowpack ($v$) and the snow height ($h$) extracted from the DSM are related as follows:

$$v = \frac{h}{TWT/2} \tag{2}$$

where $TWT$ is the two-way travel time of the wave within the snowpack in ns. $TWT$ is extracted from the radargrams by taking the difference between the air/snow interface and snow/ground interface two-way travel times.

The relative permittivity of the snowpack is a complex number. Its real part ($\varepsilon_s'$) is calculated using Neal (2004):

$$\varepsilon_s' = (c/v)^2 \,, \tag{3}$$

where $c$ stands for the velocity of light in a vacuum (taken as equal to 0.3 m/ns).

### 3.6 Drone-based GPR frequency-dependent attenuation analysis

In wet conditions, the imaginary part of the permittivity of the snow ($\varepsilon_s''$) is estimated using the GPR frequency-dependent attenuation analysis method proposed by (Bradford et al., 2009). In the standard GPR frequency range (10 MHz–1 GHz), $\varepsilon_s'$ is strongly dependent on $LWC$ and assumed to be independent of frequency. Assuming that the frequency-dependent attenuation of an electromagnetic wave through water is linearly related to frequency (Turner and Siggins, 1994), the attenuation coefficient over the GPR signal band can be written as:

$$\alpha = \alpha_0 + \frac{\sqrt{\mu_0 \varepsilon_s'}}{2Q^*} \omega \tag{4}$$

Where $Q^*$ represents the generalization of the attenuation quality parameter in the linear region of the attenuation, $\alpha_0$ the impact of low frequencies in the radar attenuation, $\omega$ is the angular frequency and $\mu_0$ the permeability in the free space.

Within the frequency range of 1 to 1500 MHz, $Q^*$ is assumed to be constant (Bradford et al., 2009) and related to $\varepsilon_s''$ as follows:

$$Q^* = \frac{\varepsilon_s'}{2\varepsilon_s''}, \tag{5}$$

Where a GPR generates waves in the form of a Ricker wavelet, the frequency $f_0$ of the spectral maximum of the GPR wave, measured at the snow/air interface on the radargram, and the frequency $f_t$ of the spectral maximum, measured at the ground/snow interface, are related to $Q^*$ (Bradford, 2007):

$$\begin{cases} \frac{1}{Q^*} = \frac{4}{TWT} \frac{\omega_0^2 - \omega_t^2}{\omega_0^2 \omega_t} \\ \omega_0 = 2\pi f_0 \\ \omega_t = 2\pi f_t \end{cases} , \tag{6}$$

In the present study, $f_0$ and $f_t$ were measured by randomly sampling at least 10 points on each GPR line. For each selected point, readings were made on five consecutive traces using the Prism2 software. Peak frequencies for each point were calculated by taking the median of the measurements. When at least one trace showed a higher frequency at the ground/snow interface than at the snow/air interface, two extra traces were used; $\varepsilon_s''$ was then computed using equations (5) and (6), with $\varepsilon_s'$ being calculated using equations (2) and (3).

$LWC$ and the relative density of dry snow ($\rho_d$) were then calculated with the following set of dimensionless empirical equations proposed by Tiuri et al. (1984) and Sihvola and Tiuri (1986):

$$\varepsilon_d' = (1 + 1.7\rho_d + 0.7\rho_d^2) \,, \tag{7}$$

$$\varepsilon_s' = (0.1LWC + 0.8LWC^2)\varepsilon_w' + \varepsilon_d', \tag{8}$$

$$\varepsilon_s'' = (0.1LWC + 0.8LWC^2)\varepsilon_w'' \,, \tag{9}$$

where $\varepsilon_d'$ is the bulk permittivity of dry snow, and $\varepsilon_w'$ and $\varepsilon_w''$ are the real and the imaginary parts of the relative permittivity of pure water, respectively.

Equations (7), (8) and (9) were established for a measurement frequency of 1 GHz and were assumed to remain valid for the purpose of the present study.

The relative snow density ($\rho$) was then calculated with the following equation:

$$\rho = \rho_d + LWC, \tag{10}$$

Finally, the *SWE* was calculated using the relative density ($\rho$) calculated using equation (10) and with the snow depth (*h*) extracted from the DSM produced by photogrammetry using equation (1). *LWC* maps were then produced by extrapolating the punctual results to areas value using an inverse distance weighting interpolation with barriers set at two metres. For each *LWC* map, 10 ± 2 points per transect were randomly selected for the interpolation. The inverse distance exponent of distance used was set at two, the maximum distance for data calculation was set at two metres, and the minimum number of points considered in the calculation was three. The interpolation was used to create *LWC* maps of 5 cm cell size.

For the surveys on February 26 and March 5, the snowpack was assumed to be dry, as the surveys were preceded by several cold days, and as the snowpack temperatures measured were all below 0°C. The *SWE* was determined using the assumptions that $\varepsilon_d'=\varepsilon_s'$, $\rho_s=\rho_d$, and *LWC*=0.

For the surveys on March 12 and March 19, the *SWE* was determined following the GPR attenuation method.

## 4 Results

### 4.1 AWS

Figure 3 presents the AWS relevant measurements over the study period. February 27 measured snowpack temperatures were all below 0°C and had not been altered by any significant ROS or major melt event yet.

Between February 27 and March 1, the snowpack was affected by a first mild episode (M.E.1) that ended with a ROS event. Mild episodes are here defined as more than 24-hour-long periods with continuous above-zero air temperatures. The first week of investigation was also characterized by several snow precipitation events. The ROS event lasted 45 minutes, with 1.6 mm of cumulative precipitations during the first 30 minutes and only 0.1 mm in the last 15 minutes. M.E. 1 warmed up the snowpack to nearly 0°C at all measured depths, generated slightly more than 1 mm of cumulative outflow at the base of the snowpack, and increased both the *SWE* and relative snow density by 15 mm equivalent and 0.05, respectively. Outflow at the snowpack's base started while the measured snowpack temperatures were still negative, suggesting that at least part of the outflow was made of liquid precipitation flowing through the snowpack. M.E. 1 was followed by a drop in air temperature of 25 degrees C, starting the beginning of a seven-day-long cold period. Over that cold period, the measured snowpack temperatures dropped below 0°C, while *SWE* and relative snow density stabilized. Outflow at the snowpack's base stopped 24 hours after the temperature started to decrease. Considering that during this mild episode at least part of the ground remained frozen, with negative temperatures observed between 0 and -30 cm under the soil surface, it is assumed that no significant ground infiltration occurred during M.E. 1 (Dingman, 1975). This suggests that most of the rain percolation either froze inside the snowpack or flowed longitudinally at its base.

The second survey occurred March 5, during the seven-day-long cold period described here above, over a dry snowpack.

A second mild episode (M.E. 2), started on March 8 and ended March 12, the day of the third survey. On March 11, the air temperature reached a maximum of 15°C and a ROS event occurred. It took two days for the warm conditions to warm the snowpack up to 0°C at all measurement points and to generate outflow at the snowpack's base. The second ROS event occurred on an already warm snowpack and produced 0.8 mm of precipitation over 30 minutes. This ROS event was therefore slightly less intense than the first one. The snow lysimeter measured a cumulative outflow comparable as during M.E. 1. As the soil remained frozen during this second mild episode, most of the water coming from the melt and from the precipitation was expected to have flowed horizontally at the base of the snowpack.

March 12 marked the last day of M.E. 2 with air temperatures falling under 0°C. Between March 8 and March 12, snow depth decreased by almost 30%, while *SWE* remained almost unchanged, despite substantial liquid precipitations being recorded. Snowpack temperatures and outflow measurements indicated that at least some of the snowpack layers were wet at the time of the survey. From March 12, air temperatures remained negative until March 17. A refreezing front slowly moved down the snowpack, and the outflows stopped over that period, suggesting a gradual drying of the snowpack. From March 17 to 18, a

third mild episode (M.E. 3) brought the snowpack temperatures back to the melting point. An outflow of minor amplitude compared to those observed during the two first mild episodes was measured on March 18 only.

During the March 19 survey, the air temperature reached a maximum of -2°C. At that date, the snowpack was drying, with an almost continuous increase of snow density and a decreasing snowpack depth. These reached 0.48 and 38 cm, respectively.

## 4.2 Drone-based photogrammetry

Snow cover maps produced using drone-based photogrammetry are presented in Figure 4. On top of the uncertainty estimation described above, the maps are fully consistent with field observations. The quality of the DSM allows for identifying specific
features such as trails used to access different features of the study area (e.g., lines in yellow on the 2020-02-26 map). Those trails show lower than pristine snow depths in a consistent way. Similarly, extra snow accumulation in drainage ditches (e.g., brown area in the bottom left area of the 2020-03-19 map) is well marked and consistently apparent on the different sub-figures). Overall, the snow depth maps are considered to have satisfactory accuracy for the purpose of the study.

By comparing the different Figure 4 maps, we can observe that the snow depth decrease that occurred between February 26
and March 5 is homogeneous over the entire area, with no differences between the flat and sloped areas being visually noticeable. Snow depth in both areas on February 26 ranged between 60 cm and 90 cm, while on March 5, the snow depth ranged from 50 cm to 80 cm in both sloped and flat areas. The situation is different when comparing the March 12 map to these two first dates. On March 12, flat area snow depth ranged from 35 cm to 55 cm, whereas the sloped area snow depth was between 25 cm and 50 cm. The severe ablation and/or settling that affected the study area impacted the sloped area more than
the flat one. Changes in snow depth were less pronounced between March 12 and 19 than for the previous periods. Maximum snow depth in the flat area decreased from 55 to 50 cm, and from 50 to 45 cm in the sloped area between March 12 and 19. Between March 5 and March 12, the maximum snow depth in the flat area decreased by 25 cm and by 30 cm in the sloped area.

Overall, Figure 4 shows that the sloped and flat sections had comparable snow depths at the end of the accumulation period
but reacted differently to ablation, with a faster loss of depth in the sloped area than in the flat one.

## 4.3 TDR monitoring

Relative permittivity measured using TDR probes and normalized to the value measured on February 26 at 12:30 are presented in Figure 5. As described in the Methods section, rapid variations in relative permittivity are associated with a change in *LWC*. Given that an increase of more than 0.1 in normalized permittivity in 15 minutes can be considered as due to a change in *LWC*,
the normalized relative permittivity is here used to assess the snowpack response to mild episodes in terms of water content and flow-through dynamics.

- M.E. 1. The first reaction to the M.E. 1 in terms of *LWC* was observed on top of the layer $\varepsilon$ (Figure 5a). That increase in LWC led to an increase in normalized relative permittivity of 0.2 on February 27, the day after M.E. 1 began. Snowpack response to the February 28 ROS event occurred first at the bottom of the sloped area, as suggested by the
increase of 1.2 of normalized relative permittivity over the $\alpha$ slope layer (Figure 4d). followed by an increase of 0.2 measured by the other TRD sensors. The detection of an increase in *LWC* at the base of the flat area occurred half a day after the increase for the sloped area, with an increase of about 0.7 of the normalized permittivity above both flat and slope $\alpha$ layers, at a time when the air temperature had already dropped below zero. Interestingly, the lysimeter measured an outflow at the base of the snowpack in the flat area 24 hours before any moisture increase was detected
by the TDR probe placed on the ground in the flat area. Differences observed in timing and amplitudes at the different probe locations suggest that liquid water flows followed preferential pathways. Past that time, normalized permittivity steadily decreased in all spots, reaching a plateau representative of the new dry densities of the snow layers. The relative permittivity of the new plateau was10% higher than on February 26, suggesting a slight increase in density.

-

-     M.E. 2. As was the case at the M.E. 1, the increase in normalizer relative permittivity was first observed at the base of the sloped area. The increase was followed by an increase at the other probes 24 hours later, those above the $\beta$ and $\gamma$ layers being of very low amplitude: 0.1 to 0.2 of normalized permittivity. The strongest increase in *LWC* was measured over the sloped ground layer (Figure 4d), with a normalized relative permittivity 4 times higher than the one measured February 26. At the end of March 10, three of the four probed layers showed an increase in *LWC*, which

was more pronounced in the sloped area than in the flat area. After March 10, the fluctuation of *LWC* above the $\gamma$ layers in the sloped area (Figure 4c) started to mimic the one above the ground, but with a lower amplitude. This synchronism suggests that the sloped area's preferential pathway flow-through mode started weakening.

    -     M.E. 3. The sloped area showed a faster and more intense response to M.E. 3 starting March 17 than the flat area. Unlike the flat area, the slope's *LWC* fluctuation started exhibiting a strong diurnal pattern, whose peak occurred a

couple of hours before the peak in air temperature and the peak in lysimeter outflow.

-

Overall, the TDR probes showed a faster and more intense response to air temperature warming episodes on the slope compared to the flat area, the presence of preferential pathways (particularly at the start of the ablation period), and a noticeably higher influence of solar radiation on the ablation of the sloped area compared to the flat one at the end of the study period.

## 375  **4.4 Drone-based GPR permittivity measurement**

On each survey day and in each area, snow depth was extracted from the DSM following the north-to-south transects (Figure 1b) covered by the M600 pro. Snow depth and snowpack bulk permittivity profiles of selected transects are shown in Figure 6.

On February 26 (Figure 6a), the flat area showed quite stable bulk permittivity and snow depth profiles, with a relative

permittivity ranging from 1.2 to 1.6. The sloped transect exhibited slightly lower snow depth and higher bulk permittivity than the sloped section, between 1.4 and 2. The bulk permittivity over the sloped section appeared more variable than the flat one too. With a relative permittivity ranging from 1.2 to 1.5 and from 1 to 2 for the flat and the sloped areas, respectively, March 5 (Figure 6b) showed limited changes in bulk permittivity compared to February 26 for both areas.

The March 12 (Figure 6c) transects showed a sharp change compared to the two first dates. Both areas exhibited a rise in bulk

permittivity and a decrease in snow depth. Bulk permittivity profiles showed gaps due to the GPR signal not penetrating fully through the wet snow. They ranged from 1.9 to 2.5 and from 2 to 3.2 for the flat and the sloped areas, respectively. The bulk permittivity in the sloped area had higher values and variability than in the flat area. On March 19 (Figure 6d), the snow depth in the flat transects remained similar to that measured on March 12. The bulk permittivity decreased to values situated between those of March 5 and 12, reaching minimal relative permittivity values of 1.5 in both flat and sloped areas, and increasing the

values ranges in both flat and sloped areas. The sloped area transect exhibited a decrease in bulk permittivity, like that of the flat area transect, and its variability remained higher than in the flat section. The main difference between the two sections was the snow depth. The sloped area showed a more pronounced decrease than the flat area. As no fresh precipitations were recorded between March 12 and 19, the decrease in permittivity in both sections can be interpreted as a decrease in *LWC*, which could have occurred together with snow densification in the sloped section.

Overall, Figure 6 confirms the difference in response to M.E. 2 between the snowpack's sloped and flat areas, including a high moisture content for both areas and a more pronounced densification of the snowpack over the sloped area compared to the flat one.

**4.5 Drone-based GPR frequency-dependent attenuation analysis**

Contradicting snow temperature profiles (Figure 2c) that suggested the snowpack was dry, the *LWC* calculation and interpolation presented in Figure 7 suggested non-zero *LWC* (ranging from 0 to 3.5%), with no visible differentiation between the sloped and flat areas on both February 26 (Figure 7a) and March 5 (Figure 7b). On both dates, there was a relative spatial heterogeneity in *LWC*, with no common patterns between the two dates.

On March 12 (Figure 7c), the flat area *LWC* ranged between 0 and 5.5%, while the sloped area had *LWC* maximal values above 8%. The March 12 survey showed a general increase in *LWC* compared with the two previous surveys, and a differentiation between the two studied areas. The sloped area exhibited the highest overall LWC, although both areas were spatially variable. Compared to March 12, March 19 (Figure 7d) showed overall slightly lower *LWC* values in the sloped area compared to the flat area. However, the maximum value reached 6.5% in both areas, making them difficult to differentiate. *LWC* values remained highly variable for both sections, ranging between 0 and 6.5%.

Overall, Figure 7 confirms that, unlike the ROS event that occurred at the end of February, the sloped and flat areas responded in different ways to the March 11 ROS event. On the other hand, *LWC* values seem unrealistic for the two first survey dates that followed the pronounced cold episode. In a similar way, the absence of a recurrent spatial pattern in *LWC* variations between maps of different dates suggests the method was not able to capture these variations in a detailed way.

**5 Discussion**

Drone-based estimation of key snowpack variables

The spatiotemporal variability in snow depth, snow density, *SWE* and snow *LWC*, four key properties of a snowpack, has been assessed using drone-based GPR and photogrammetry methods in a repeated way. Figure 8 provides an overview of the variability of those properties in the form of boxplots and, where possible, compares drone-based measurements to those of the AWS and from snow pits.

Photogrammetry snow depth results (Figure 8a) are in good agreement with those of the AWS and of the snow pits over the entire study period, with a possible slight overestimation in the two first surveys. The differences between the 25th and 75th percentiles in the flat area are systematically below 2 cm. For comparison, this difference is of the same order of magnitude as the one between the snow pit and AWS measurements and the median. The slope is characterized by a lower snow depth and a larger range than the flat area, especially after the ROS event that occurred on March 11. With most of the slope snow depth values below 40 cm the two last surveys, the estimated ±5 cm uncertainty that applies to the photogrammetry affects more than 12% of the measurement. Such high level of uncertainty may have potential detrimental effects on the GPR-based calculation of key snowpack properties.

The *LWC* boxplot in Figure 8b is effective in representing the general evolution of the snowpack moisture content through time: a stable situation occurring between the first two survey dates, followed by a marked increase in snow moisture on March 12 and a slight decrease on March 19 (for the sloped area only). The boxplot also successfully captures the difference in response to mild events between the flat and sloped areas. Compared to the A2 WISe sensor measurements, the boxplot shows the method did not succeed in providing realistic *LWC* values. According to the A2 measurements, snow pit bulk *LWC* values were close to 0% February 26, March 5 and March 12. while the GPR-based calculation medians for the flat area were 2, 1.5 and 4%, respectively. The differences between the 25th and 75th percentiles in the flat area were 1, 1.5 and 2% for the same dates. Even if the A2 measurements might have been influenced by the sampling constrains and therefore might have underestimated the snowpack average *LWC*, the drone-based result appears significantly overestimated.

Disagreement between GPR based calculations and reference measurements were observed for the relative snow density as well (Figure 8c). February 26 and March 5, the difference between reference values and GPR based calculation medians was 3 times higher than the difference between the 25th and 75th percentiles. The drone-based method underestimated relative snow

density over the flat area compared with both the AWS and the snow pit measurements for the first two and the last dates, while overestimating it for March 12. Moreover, both the flat and sloped areas exhibited an unrealistic 50% decrease in snow relative density between March 12 and 18. No fresh snowfall occurred between those two dates.

Calculated as the product of $h$ by $\rho$, *SWE* boxes show similar biases as relative snow density (Figure 8d).

Interestingly, we note that while Figure 6 shows the bulk permittivity profile being consistent with TDR and AWS measurements, this is not the case with the GPR-based computed variables presented in Figure 8. As described earlier, the bulk permittivity of the snowpack is influenced by both snow density and *LWC*. Figure 8 therefore suggests that the method we applied failed to differentiate the relative influence of both variables.

The method we applied makes use of empirical equations (7), (8) and (9), which are commonly used in snow hydrology. According to Tiuri et al. (1984), Equations (8) and (9) apply to pendular regime, for $\varepsilon_s'\leq2.6$ ($\varepsilon_s'\leq3$ for Colbeck (1982)), as opposed to a funicular regime. In a layered snowpack in which preferential flow occurs, it is realistic to hypothesize that both regimes occur in the snow column, making Equations (8) and (9) possibly not directly applicable to bulk relative permittivity measurements.

Different empirical formulas relating the relative permittivity to relative snow density have been subsequently developed (e.g. Di Paolo et al. (2018); Frolov and Macheret (1999)). They could possibly represent more accurate alternatives.

As suggested by Webb et al. (2021), reassessing the application conditions of the equation used in the present study is another direction that could be chosen. Selecting different equations depending on snowpack conditions and evolution over the winter could ensure a better use of these equations too.

Fixing the relative density of the snow based on manual sampling or AWS values could represent another solution to the problem encountered in differentiating between the relative influence of snow density and *LWC* on bulk permittivity. However, this solution would not allow for the capturing of the spatial variability in snow density, and therefore might bias calculations.

Spatiotemporal variability in snowpack characteristics. TDR monitoring, drone-based photogrammetry and drone-based GPR have been shown to be a valuable combination for assessing the spatiotemporal variability in key snowpack variables. The use of photogrammetry to map snow depth over the study area provided the opportunity to calculate bulk permittivity from repeated drone-based GPR surveys. Both bulk permittivity and snow depth profiles agreed with site observations and reference measurements. When the bulk permittivity was converted into absolute snow density, *LWC* and *SWE* values did not provide the expected results even if the temporal evolution of those parameters was captured in an acceptable way. TDR monitoring complemented the drone-based measurements well, providing both high temporal resolution and layer-based snowpack relative permittivity time series. Snow depth and snow bulk permittivity calculations were highly consistent in comparisons of the different methods to each other, allowing for the capture of the flat and sloped areas responses to changes in meteorological conditions.

Points learned from the case study

The application of the proposed methodology to the winter 2020-21 led to the following facts being learned:

- The flat and sloped areas had comparable responses to the first ROS event of the study period, which occurred on a cold and dry snowpack at the end of February. That event produced snowpack outflows and increases in *LWC*, especially at the base of both areas. The sloped area, however, showed a faster and more intense response than the flat one.
- The first ROS episode did not modify the snowpack's snow density and snow depth profiles in a substantial way. Both study areas exhibited characteristics of preferential flow pathways.
- The second ROS event that occurred on March 10 on an already pre-warmed snowpack affected the sloped area in a different way than the flat one, both areas showing important differences in snow depth, *LWC* and density in the

March 12 surveys. The timing and amplitude of the outflow suggest a more homogeneous flow path was present than during the first ROS.

- The third mild episode that occurred from March 16 to 18 did not drastically modify the characteristics of either area compared to the March 12 situation. However, the slope showed faster rates of melt/ablation and showed higher response to diurnal fluctuations, probably due to its southerly aspect.

## 6 Conclusion

A combination of TDR monitoring, drone-based photogrammetry and drone-based GPR was used in the experimental watershed of Ste-Marthe (Quebec, Canada) over the winter of 2020–2021. The suite of methods showed comparable snow accumulation over flat and sloped areas, with comparable characteristics lasting after the first ROS event. The second ROS event at the start of the ablation season led to differences in response between the two areas.

Drone-based GPR was very instructive when interpretation was based on bulk permittivity results, but showed limitations in mapping snow density, *SWE* and *LWC*. There are questions about the applicability of empirical equations used given the site conditions. The results suggest the empirical equations should be reassessed for conditions that differ from the ones for which they were formulated. The method did not allow the researchers to obtain the full benefit from applying the GPR frequency-dependent attenuation method to estimate *LWC* in the snowpack. The method shows promise, however. In the winter of 2020-2021, the radargram obtained using a 1.5 GHz GPR was not detailed enough to differentiate between the main snowpack layers. However, efforts should be continued in this regard, as the 2020–2021 snowpack was characterized by a relatively low snow depth and an uneven distribution of the ice layers in the snow column.

## Author contributions

MB, ER and FB framed the research project, determined the objectives and made up the research project steering group. EV designed the research and organized the fieldwork. EV and MB collected data in the field. CM conducted a substantial part of the GPR data treatment. EV produced all the other results and made the interpretation. EV wrote the initial draft of the paper. All authors contributed to editing and revising the paper.

## Code/Data availability

The data are available on request to the corresponding author.

## Competing interests

The authors declare that they have no conflict of interest.

## Acknowledgment

This research is supported by the Geochemistry and Geodynamics Research Centre (Geotop) of Quebec; the Quebec Water Research Center (Centreau); FRQNT and the Natural Sciences and Engineering Research Council of Canada (NSERC); and the Ecole de Technologie Superieure, a member of the Université du Quebec. The authors are grateful for the invaluable support of the municipality of Sainte-Marthe, Quebec, Canada.

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

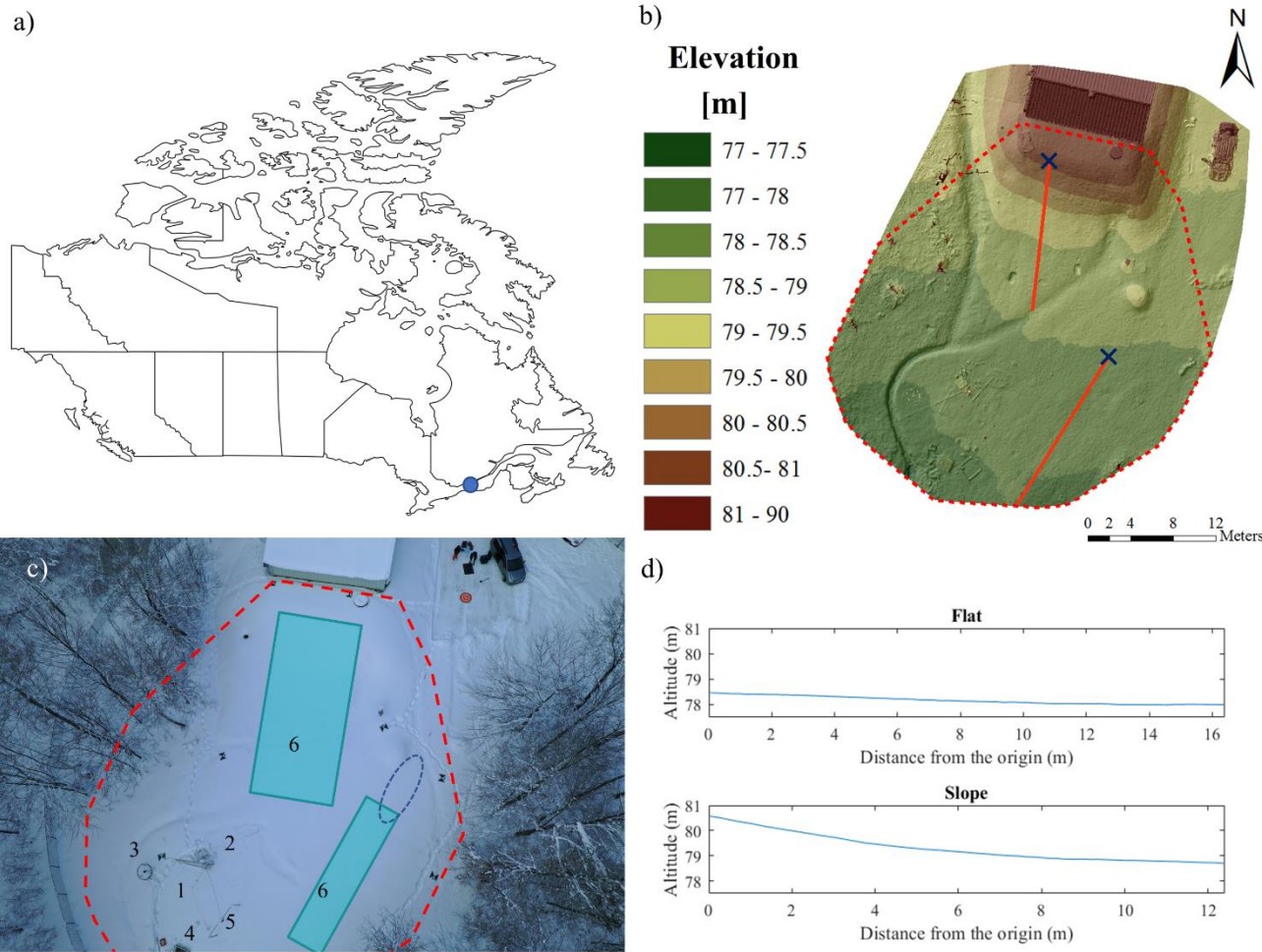

**Figure 1: Study Site. a)** Location of the BVE Ste-Marthe. **b)** Snow-free DSM of the main station area; red polygon delimits the study area, red lines represent the two studied transects and blue cross mark the two profiles' origins. **c)** Overview of the BVE Ste-Marthe main station; red polygon delimits the study area, blue areas represent the two studied areas and dashed dark blue ellipses represent the zone used for the snow pit. Numbers identify devices of interest for the present study: **(1)** sonic sensor, **(2)** ground and snow temperature sensors, **(3)** shielded precipitation gauge, **(4)** snow lysimeter, **(5)** *SWE* sensor and **(6)** TDRs. **d)** Altitude profile of the two studied transects.


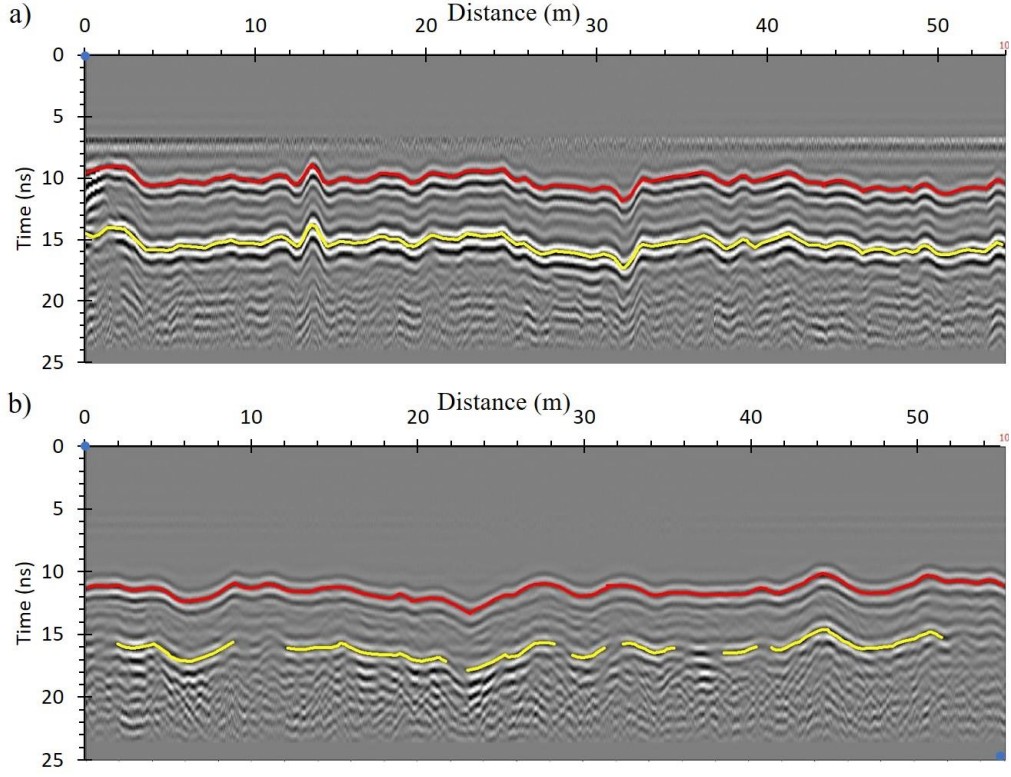


**Figure 2: Flat area drone-based 1.5 GHz GPR radargram collected on a) March 5 and b) March 12 (considered as the less legible radargram). The red line represents the air-snow interface and the yellow line represents the snow-ground interface.**

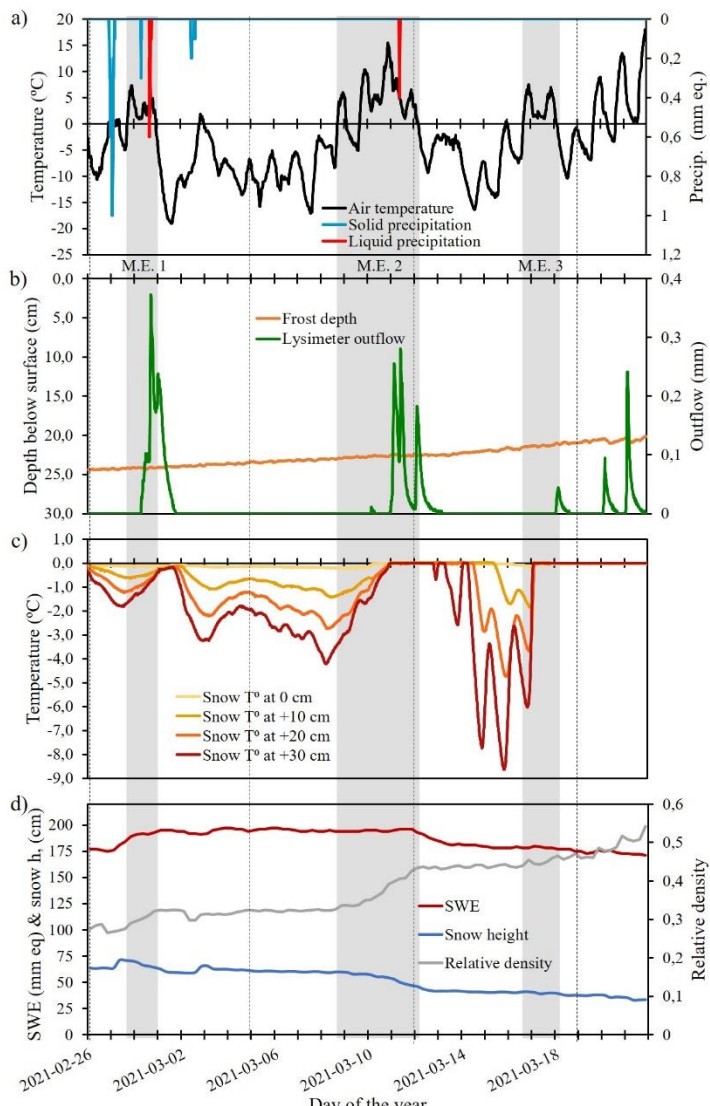

**Figure 3: AWS measurements during the winter 2022 ablation period. a) Air temperature and precipitations; b) frost depth and lysimeter outflow; c) snow temperature at four different heights; d) snow water equivalent, snow height and relative snow density. Variables associated to the different line colours are indicated in each subfigure. Semi-transparent grey shadings represent mild episodes. Mild episodes identification is given underneath subfigure a). Details about variable descriptions and measurements are given in Table 1. Vertical dashed lines indicate field measurements days.**

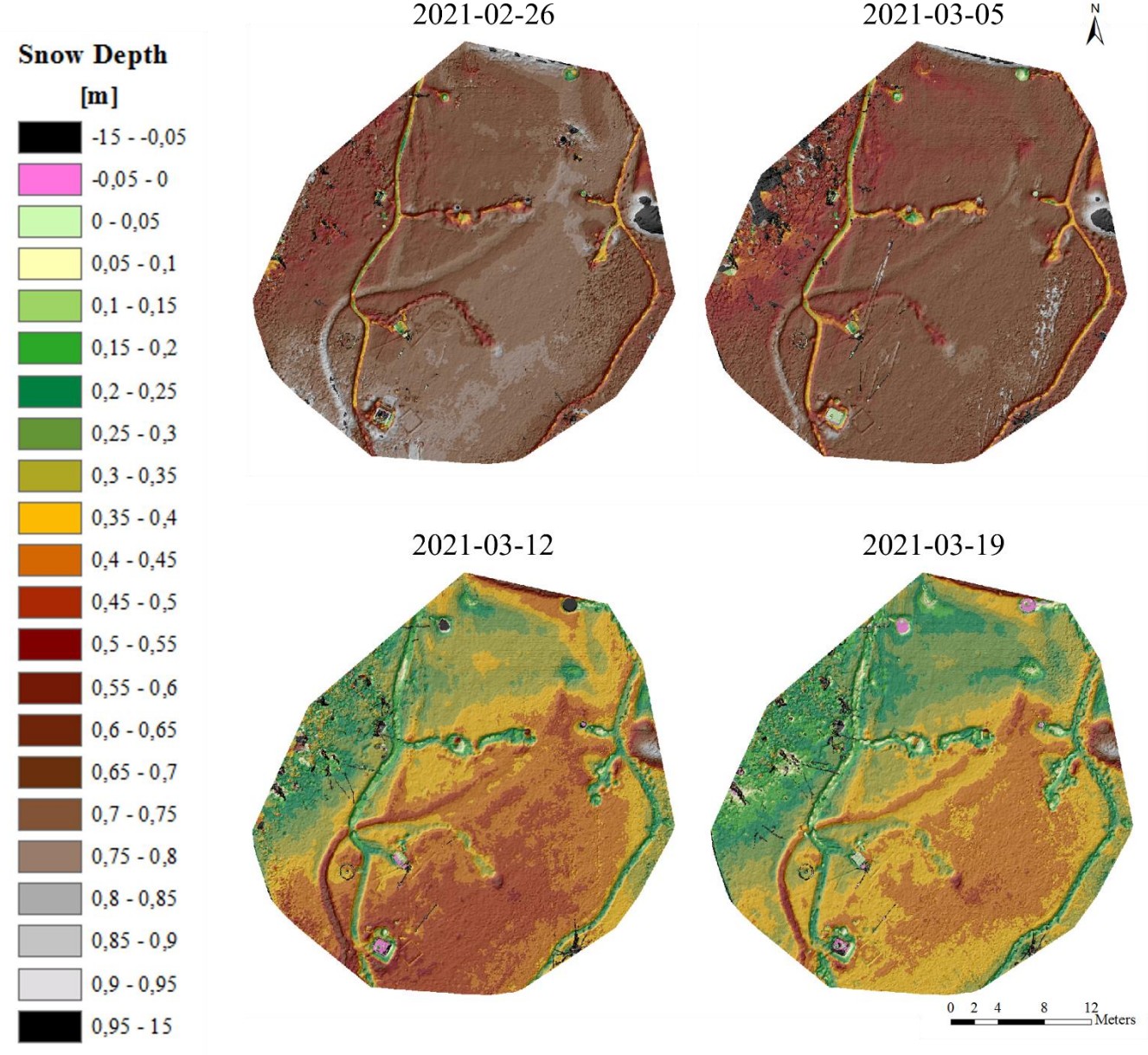

**Figure 4: Snow depth calculated by photogrammetry for the four dates covered in this study.**

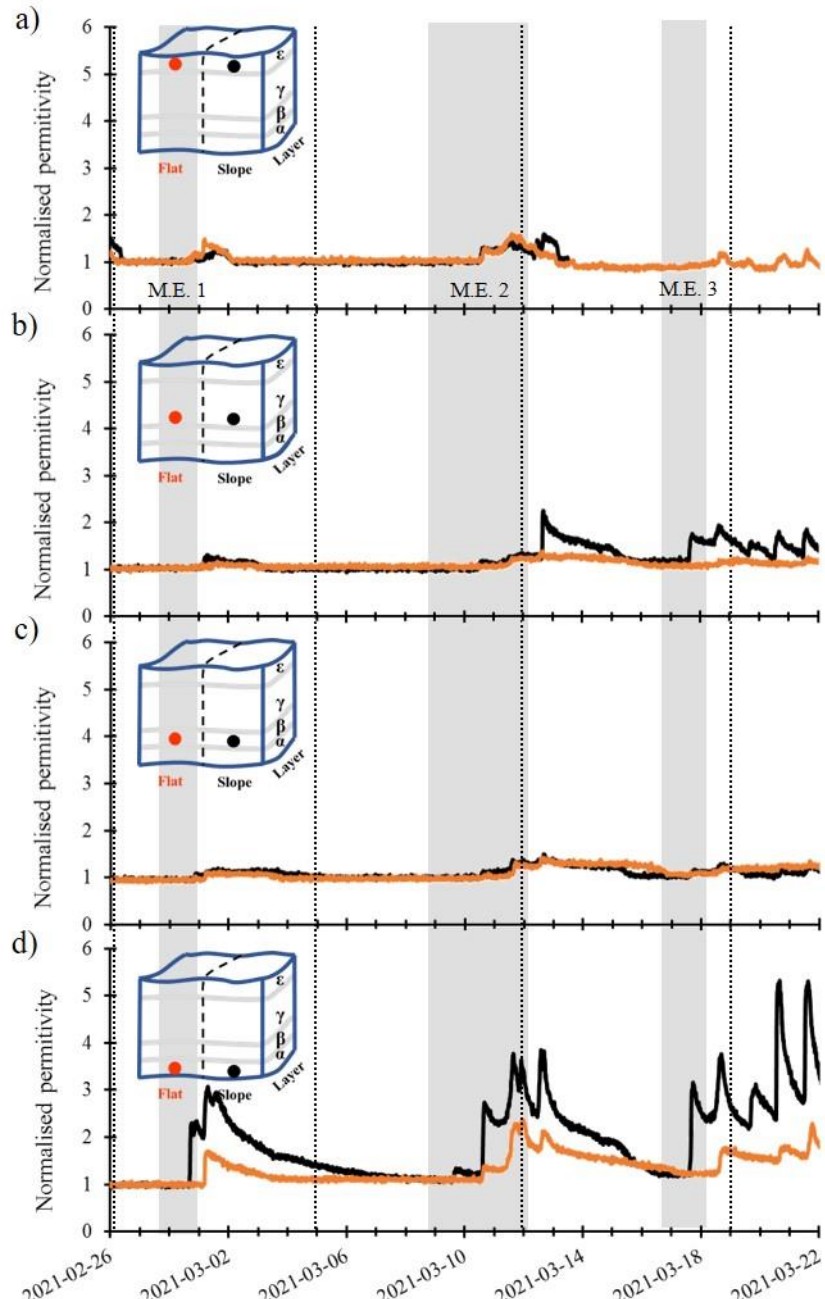

**Figure 5: Normalized permittivity measured by TDR probes in the sloped (black line) and flat sections (orange line).**
**Probe positions in each graph are shown in drawings representing a simplified description of the snowpack, with layer**
**identification letters: a) layer ε; b) layer γ; c) layer β and d) layer α. Semi-transparent grey shadings represent mild**
**episodes. Mild episodes are identified in sub-figure a). Vertical dashed lines mark field visit dates.**

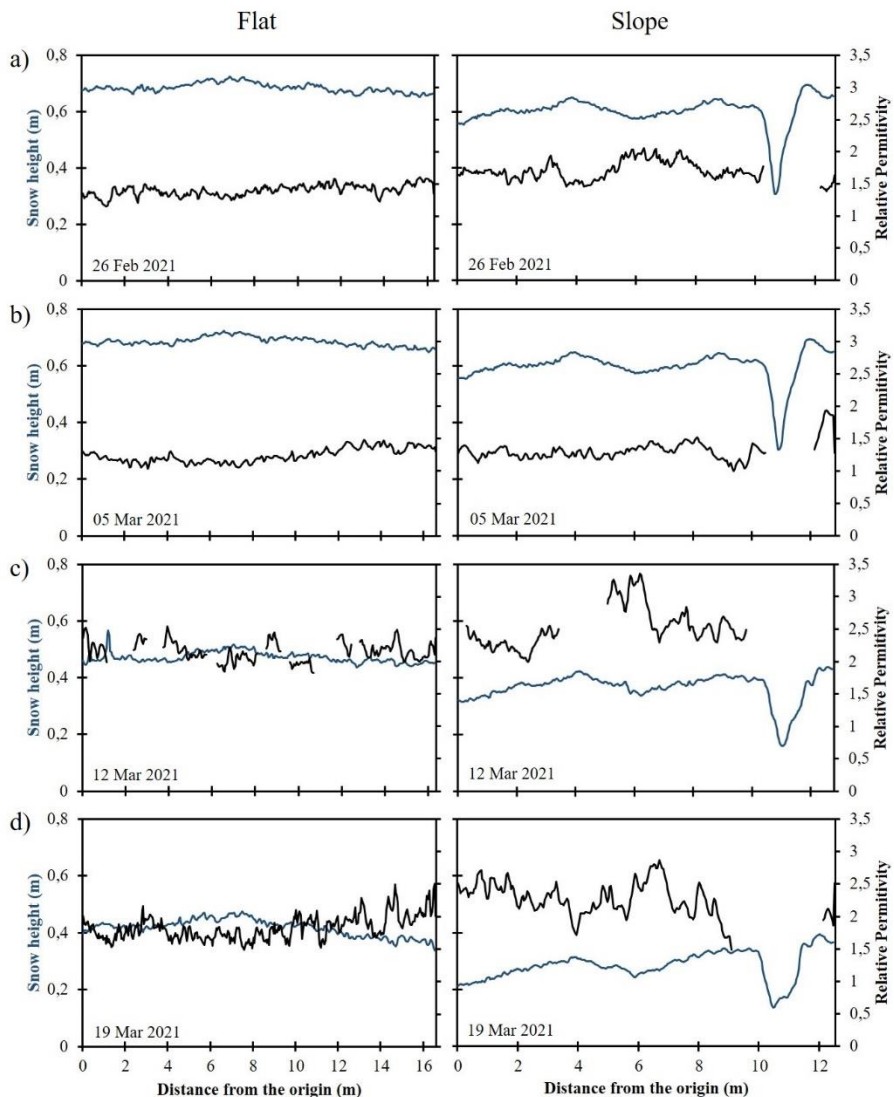

**Figure 6: Bulk permittivity (black line) and snow depth (blue line) calculated for the flat (left) and sloped (right) transects on: a) February 26; b) March 5; c) March 12; and d) March 19. Adapted from Valence and Baraer (2021).**

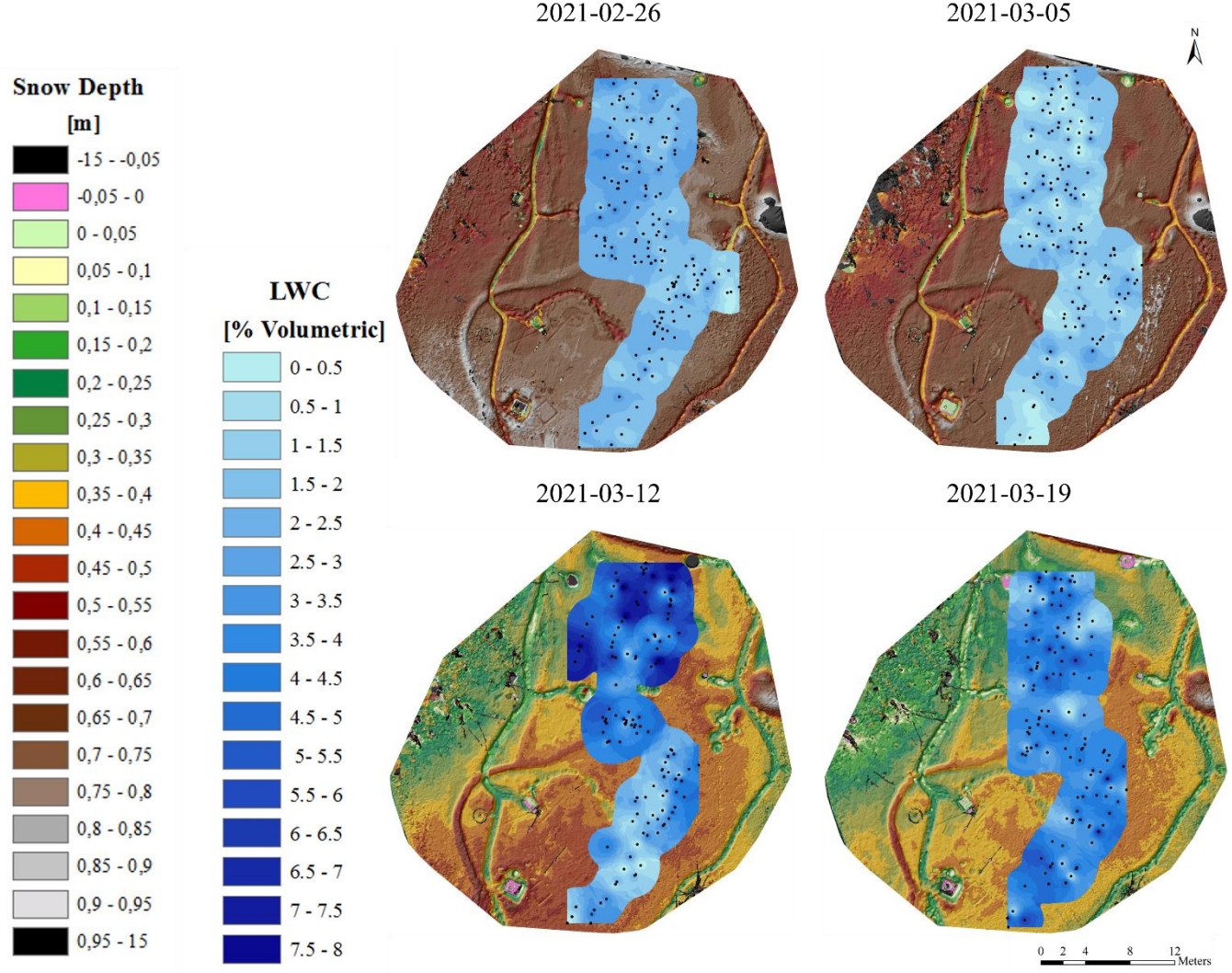

**Figure 7: LWC calculated by GPR frequency dependent attenuation analysis. Points used for interpolation are displayed in black.**

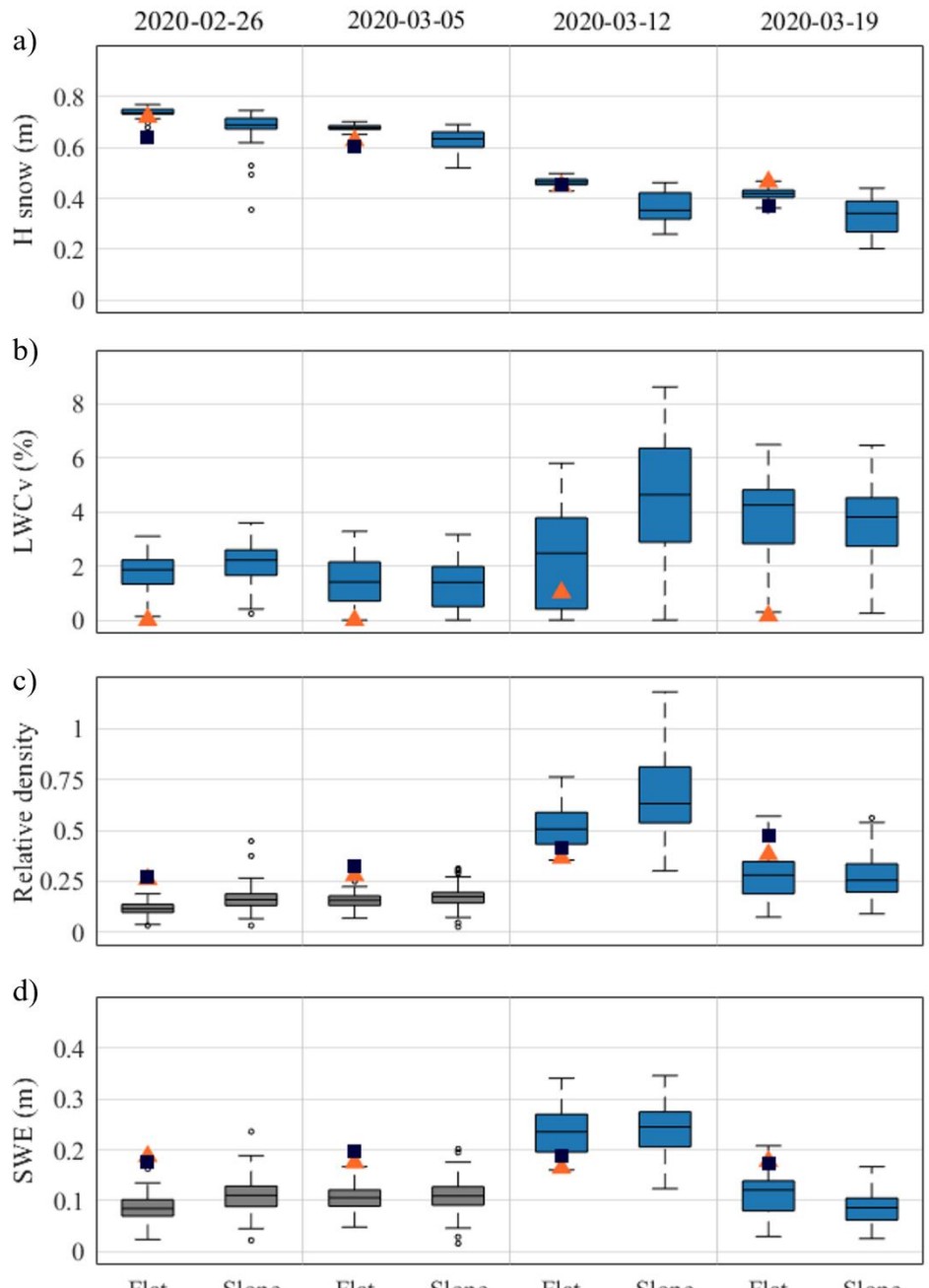

**Figure 8: Box plots representing the snowpack studied variables for the sloped and flat areas for each survey date: a) Snow depth results from photogrammetry; b) Liquid water content estimated with drone-based GPR; c) Relative density estimated with drone-based GPR; and d) Snow water equivalent calculated from relative density and snow depth. In the boxes, the central black line represents the median, and the bottom and top edges mark the 25th and 75th percentiles, respectively. The whiskers show the data ranges, excluding outliers. Where they exist, outliers are represented by a black circle. Dark blue squared and orange triangles markers represent reference values originating from the AWS and snow pits, respectively. Grey boxes were calculated with the assumption of a dry snowpack.**

**Table 1: List of the instruments used in this study. Accuracy is either given by the manufacturer or estimated for worst-case scenarios.**

**Adapted from Paquotte and Baraer (2022).**

| Variable | Sensors | Manufacturer | Accuracy | Timestamp |
|----------|---------|--------------|----------|-----------|
| $T_{air}$ | Hygrovue10 | Campbell Scientific | ±0.6°C | 15 min |
| Precipitation | Tipping Bucket Rain Gauge (52202) | Campbell Scientific | ±3% | 15 min |
| Snow depth | Ultrasonic (SR50A) | Campbell Scientific | ±1 cm | 15 min |
| $T_{n,cm}$ | Thermal profiler (CS230) | Campbell Scientific | ±0.2°C | 15 min |
| Outflow | Lysimeter | Homemade | ±1% | 15 min |
| $SWE$ | $SWE$ sensor (CS725) | Campbell Scientific | ±15 mm | 6 h |
| Permittivity $(\varepsilon)$ | TDR (CS610) | Campbell Scientific | ±5% | 15 min |

**Table 2: Methods combined in this study and classified based on the sampling frequency and the spatial coverage.**

| Variable | Continuous, single point | Repeated, single point | Repetitive, two surfaces |
|----------|--------------------------|------------------------|--------------------------|
| $h$ | Sonic sensor | Snow pit | Photogrammetry |
| $\rho$ | $h / SWE$ | Snow pit | GPR |
| $SWE$ | $SWE$ sensor | Snow pit | GPR |
| $LWC$ | TDR (2 points, 4 layers) | A2 | GPR |