# Peer review of "Drone-based GPR application to snow hydrology"

_The Cryosphere, 2022_

## Author Comment (AC2)

The authors would like first to thank both reviewers for their detailed and pertinent comments. There is no doubt that those comments will serve to improve the manuscript substantialy.

**RC1**

**Summary:**
Valence et al. present findings from a field campaign in Quebec in 2020-21 that integrates novel drone-based GPR snow property retrievals and comprehensive in situ observations from both automated instruments and snow pit observations. These methods are quite new and exciting, and I expect that this paper will be of significant interest to the research community. Below, I provide my general and specific comments.

**General comments**

1. Quantify findings and include these details in the abstract

At multiple locations in the manuscript (noted in Specific Comments), changes in the snowpack characteristics (the results) are presented in a general, non-quantitative manner; e.g. "the LWC increased." The manuscript would be strengthened by replacing these statements with specific quantitative results from the analysis.

> We have reviewed all the specific comments detailed hereunder and corrected the manuscript as suggested. In addition, We made a review of the document to make sure all descriptions of results are now made in a quantitative way wherever possible.

The most important of these findings should be added to the abstract to replace the current focus on the TDR results, as the novel drone-based GPR results should be highlighted more prominently.

> We fully agree with this comment as well. Therefore, we have rewritten more than 35 % of the abstract to correct that aspect.

2. Refine writing

Introduction: The introduction could be strengthened by refining the writing to provide a more detailed and direct introduction to the novel work presented in the manuscript. By doing so, there would be more space for a more thorough literature review, for which I have provided a few examples in the Specific comments.

> Thanks for providing references that we missed in the previous version of the manuscript. A new paragraph has been introduced in the introduction. It includes the reference suggested hereabove and new publications on drone-based GPR snowpack measurements.

Split study and condition section: The condition details are results (and even extends into interpretation/discussion) and thus seem better suited to appear at the beginning of the results section after the Method sections introduces the AWS observations.

> The condition section has been spitted into three parts. One part was moved to the method, one to the results and one to the discussion sections. In addition, precisions regarding the AWS have been introduced in the manuscript (method and results sections).

3. Details on SfM snow depth acquisition, processing, and interpretation:

Provide more specific details in Section 3.3, for instance % overlap of images, flight pattern, UAV height AGL, processing/filtering steps, the resolution of the gridded DSM rasters, the specifics of the ground control points, the nature of the ground surface in the snow-off DSM, etc.

> Specific details, such as images overlap, flight plans and altitude, number of images, drone speed, and expected horizontal resolution, have been added in the new version of the manuscript.

Were the ground control points used in DSM model development or used as independent checkpoints? What is the basis for the reported 3-5 cm uncertainty?

> This part of the manuscript has been rewritten. DSM have first been generated without including the GCP and then corrected based on GCP. The correction was of a couple of centimetres moist of the case. The maximum correction applied was 5cm. Those results were considered satisfactory compared to what is presented in the literature and acceptable for the study.

Were the surveys done in an identical manner on all dates? Were the GCPs deployed and surveyed each time?

> Yes. GCPs were placed approximately at the same position for each survey. GCP Coordinate were recorded using the KlauPPK to perform a final DSM correction.

On what basis were the maps considered "satisfactory" (ln 273)? Were they independently evaluated by distributed snow probe observations?

> At that stage of the manuscript, the objective was to inspect the maps produced by photogrammetry visually. Comparisons to field measurements of snow depth are described at the end of the manuscript in figure 7, which presents other key snowpack characteristics. The position of fixed features and marks on the different maps was evaluated and commented. This is now clarified in the manuscript.

How were the drone-based snow depths and radar travel times aligned/integrated (e.g., Figure 5)?

> GPR measurements were geolocalized using PPK, and results were produced in the form of georeferenced maps. Snow depth transects were extracted from the snow depth maps described here above. Both maps were treated using ArcGIS to guarantee data superposition. This is now explained in the manuscript.

What was the "good agreement" between the SfM depths and those of the AWS (ln 347)?

> Figure 7a presents extracts from snow depth maps for two sections, one in the flat and one in the sloped areas. The AWS snow depth sensor is situated in the flat area at +/- 10m from the selected flat area transect. Sun exposition is comparable between the AWS sensor footprint and the flat transect. Therefore, it is expected that both snow depth records exhibit comparable results. This is now explained in the method section. In addition, a comparison of snow pits, AWS and drone-based measurements has been introduced in the discussion section.

**Specific comments**

Title: The title should be modified to more accurately reflect the key findings of the work; previous publications have used drone-based GPR, so the use of "Introducing" is not fully justified

> The title has been changed to "Drone-based GPR application in snow hydrology studies."

10 – …dictate the quantify "and timing"…. Suggested addition in parentheses
> Done.

13 – what is a mild episode?
> We named "mild episodes" short periods with air temperature close to or higher than 0 C. This is now explained in the manuscript.

13 – This study develops… delete "aims to"
> Done.

14 – replace repetitive with repeated; here and elsewhere in the manuscript
> Done.

15 – define GPR at first usage
> Done.

18 – "on a weekly basis" – while the four surveys occurred at a weekly interval, it would best to describe the scope more specifically by stating the total number of surveys, as this could be misinterpreted as surveys occurring weekly throughout the winter
> Done.

20 – properties "were" monitored using TDR…
> Done.

22 – I would encourage a revision of the abstract to focus on the main findings of this work. At present, the findings listed after "Among others…" focus on results from the TDR probes rather than the drone-based GPR work. Further, the last point "the hydrological influence…" is an interpretation based on the TDR and lysimeters, rather than a direct finding. I suggest revising to highlight the main findings from the novel GPR methods.

We have rewritten more than 35 % of the abstract to correct that situation.

33 – "reported from cold regions." I wouldn't describe all of these locations as cold regions, so I suggest describing them in a different manner.

Done, "cold regions" has been removed

39 – Specify the period of time over which the Li et al., 2019 study documents an increase in ROS events.

Done.

50 – consider replacing "capture" with more precise terminology

Done.

55 - SfM derived snow depths do not have cm scale accuracy – is this statement referring to the spatial resolution? Most previous studies document RMSEs of ~10 cm.

Bühler et al., 2016 situates snow depth measurements accuracy between 0.07 and 0.10 m. Yildiz et al. (2021) and Avanzi et al., 2018 accuracy estimates are 0.06 and 0.03 m, respectively. We therefore adjusted our text as follows "With a vertical accuracy higher than 10 centimetres, this technique allows a non-destructive monitoring of the spatial variability of snow depth (Bühler et al., 2016a; Avanzi et al., 2018)".

65 – Consider adding Holbrook et al., 2016 (Estimating snow water equivalent over long mountain transects using snowmobile mounted ground-penetrating radar) in *Geophysics*, doi:10.1190/GE02015-0121.1.

Done.

80 – add Guneriussen et al. (2001; InSAR for estimation of changes in snow water equivalent of dry snow, *IEEE*, 39(10)) reference to Rott et al., 2003

Done.

88 – add Webb et al., 2021 reference to Mavrovic et al., 2020

Done.

97 – I thought Yildiz et al. (2021) was working in dry snow conditions?

Our mistake. Our intention was to cite Lundberg et al. (2013)… This has been changed in the new version of the manuscript.

105 – add Prager et al., 2021 (Snow Depth Retrieval with an autonomous UAV-mounted Software-defined radar, *IEEE*) reference to Jenssen and Jacobsen, 2020

Done.

107 – what was the approximate area of the study plots?

This information is now added to the study site section. The areas are 30 m$^2$ for the flat and 50 m$^2$ for the slope areas.

110 – what is BVE?
BVE stands for Bassin Versant Experimental. As BVE Ste Marthe is the name of the research site we suggest leaving as it is.

111 – List the figure references as Fig. 1a. rather than Figure 1.a
Done.

112 – replace topography with topographic
Done.

113 – what is the meaning of clearance in this context?
Sorry, our mistake. We meant "clearing". Changed in the manuscript

113 – given the prominence of the slope comparisons in the findings, I would consider adding a subplot showing surface slopes to Figure 1. What is the variability in slope within these plots?
Done.

113 – delete "main station hosts"
Done.

114 – what are the hydroclimatic variables that are measured?
The sentence has been changed to integrate the list of variables used in this study (in Table 1).

118 – replace hyphens with en dashes; here and elsewhere
Done.

121-123 – given the importance of these ROS events to the findings, I suggest describing them in more detail. For instance, how did the precipitation rates vary? Duration? Cumulative amount?
This is now done and detailed in the AWS method section.

128 – what defines a "mild weather episode"? Please describe more specifically.
Done, see the answer to line 13 comment

130 – here is one example where additional specifics could be included – how much did SWE and density increase by?
This has been added to the AWS results section and commented in the discussion section.

131 – , suggesting the presence of a preferential flow path…. This is an interpretation of the results and is better suited for the discussion
A sub-section on this topic has been added to the discussion in the new version of the manuscript.

134 – I find the discussion of the weekly field observations and the "seven-day-long cold period" (which isn't aligned with the weekly field observations) to be slightly confusing. I would consider adding semi-transparent shading to define the various warm/cold intervals in Figure 2 to help guide the reader.

> Adding cold and warm periods indications to Figure 2 definitively helps the reading of the section. Mild episodes are now numbered. We have modified the text accordingly.

140 – Please define magnitude of "substantial liquid precipitations."
> This is now detailed in the manuscript.

148 – is the statement "at least part of the ground remained frozen" based on the frost depths shown in Figure 2? Clarify this and include the appropriate figure reference.
> Indeed, Figure 2b shows that, at the AWS, the soil is at a negative temperature at -20 cm and positive at -30 cm. We realize that we moved too fast from the results to interpretation without explaining our point. Therefore, precisions are now provided in that section.

160-165 – what was the sequence of these observations? How was the pit oriented?
> Snow pits were dug at every field visit at least once a week. Each snow pit was north-oriented at a spot situated at approximately 75 cm south of the previous one. After measurements, the pit was refilled with snow to avoid topographic disturbances in the area. Details on snow pit operations and WISe measurement have been added to the method section.

165 – What is meant by "Punctual"?
> Sorry, a lazy translation of the French word Ponctuel. We rephrased the sentence.

165 – what was the vertical spacing between observations in the two vertical profiles?
> Measurements in each profile were spaced 10 cm vertically. We have added that information to the manuscript

169 – replace length with thickness.
> Done.

191 – it is fairly non-traditional (in my experience) to list the manufacturer in parentheses after a specific product name. I suggest "A DJI Mavic 2 Pro UAV…"; here and elsewhere.
> Done.

204 – Provide additional details on GPR integration and flight control software, as "supplied by SPH Engineering" doesn't provide the reader with the necessary detail.
> Yes, details were missing on that aspect. We have now further explained that UgCS SkyHub onboard computer create the link between the DJI M600 Pro flight controller and the GPR. SkyHub provides precise altitude measurements from a radar altimeter to maintain the flight at the targeted elevation above the ground surface. In addition, SkyHub controls the GPR, records measurements and flight information. We have modified the manuscript to include those precisions.

215 – replace electronic with electromagnetic
> Done.

221– check equation – should this be $(c/v)_2$?

*Sorry, our mistake. This is now corrected*

230 – replace representing with represents
*Done.*

254 and 256: provide units for equations 9 and 10
*All terms in equation 9 are dimensionless: $\rho$ and $\rho_d$ are relative density (density divided by water density); LWC, the liquid water content represent a volume of water divided by a volume of snow same as Wv in Tuiri et al. (1984). In equation 10: $\rho$ is still the relative density (therefore dimensionless); SWE and h are both in meters.*

265 – see general comment #3 – were the control points used in the DSM model generation or were they completely different? What are you defining "significant differences" as?
*This part of the manuscript has been rewritten. DSM have first been generated without including the GCP and then corrected based on GCP. The correction was of a couple of centimetres moist of the case. The maximum correction applied was 5cm. Those results were considered satisfactory compared to what is presented in the literature and acceptable for the study.*

268 – replace devices with solutions
*Done.*

269 – is this sentence necessary or could it be combined with the subsequent statement?
*Done.*

276, 279 – how much did the snow depth change by?
*The Min-Max values are now given with the survey's result. In addition, the boxplot in Figure 7 now includes snow pit derived measurements and AWS measurements to allow comparison with the drone-based measurements.*

282 – given the elapsed interval, would it be better to report the magnitude of the change rather than the rate?
*As for the previous comment, this was moved to the discussion.*

285 – replace fast with rapid; how is this defined? Provide specifics.
*The section on TDR measurements in the method section has been rewritten. It now presents how we differentiate between settling and wetting influences on snow permittivity.*

286 – Consider different phrasing than "Timewise"
*Done.*

287 – Quantify the tiny increase.
*The TDR value changes are now quantitively described in the TDR results section.*

289 – what was the spatial offset of the lysimeter and the TDR probe?

The spatial offset is less than 10 m. Moreover, the lysimeter situates in the flat area of the study site and presents a comparable exposition to sunlight. This is now precised in the method section.

298 – Quantify the increase in LWC
Done

304 – the statement "This suggests the slope was more responsive…." is better suited for the discussion.
Several parts of the results section, including the one described in the comment, have been moved to the discussion.

307 – ablation rather than ablations
Done.

311 – quantify the variability in the bulk permittivity; how is "quite stable" defined"
As for the snow depth, min-Max values for the bulk permittivity are now provided in the result section. This allows for providing a quantitative description of the parameter variability.

317 – how was the bulk permittivity variability calculated?
The bulk "permittivity variability" term was removed and replaced by the range for each transect.

328 – revise the sentence to include the results rather than just stating that the results are found in Figure 6
Done.

333 – quantify the general increase in LWC
We proceeded similarly as for snow depth and the bulk permittivity. Min-Max values for the LWC are now provided in the result section. This allows for a quantitative description of the parameter variability.

335 – quantify the "highly variable" LWC for both sections.
The variability in results is now expressed using the range.

356 – how much do the drone-based results overestimate the WISe values?
Both methods do not measure precisely the same variable and do not strictly overlap. However, we believe that the comparison still provides valuable information. WISe values are now visually compared to the median and quartiles of the drone base results in Figure 7. The text in the discussion has been modified accordingly.

358 – how are the drone based estimates calculated? Mean of all observations? Median? Proximal to the pit?
A solution similar to the one presented above has been implemented in the discussion section.

387 – "except for the density, LWC, and SWE…" – I would consider rephrasing to state where there was agreement (depth, permittivity) and where there wasn't agreement.
Done.

400 – replace southward orientation with southerly aspect
Done.

414 – replace disposition with distribution
Done.

Figure 1b) add scale to figure? Is this a single image or could an orthomosaic be included instead?
As Figure 1b. is a picture, we decided to leave it without scale and North direction. However, those have been added to the Figure 1c) representing the study area map and positioned just below Figure 1b)

Figure 1c) is this the snow-off or snow-on DSM? What do the red contours correspond to?
Figure 1 has been modified, and contour lines have been removed. In addition, the snow-off conditions are now mentioned in the caption.

Figure 2 – add a, b, c and d labels to subplots. Does the RHS y-axis need to be divided by 15 in 2a and 2b?
Modifications have been made to Figure 2.

Figure 2b – is the LHS y-axis depth below surface?
Yes, Figure 2 has been modified on that aspect too.

Figure 2d – typo in snow height in legend, how was relative density calculated? Units?
The relative density is calculated using equation 10. This has been added in the AWS method section. Relative density is dimensionless.

Figure 3 – replace commas with periods
Done.

Figure 3 – Switch DD/MM date format to YYYY-MM-DD to match figures 2 and 4.
Done.

Figure 4 – add a, b, c and d labels to subplots. Is it standard for these to be normalized by the first reading? How should the reader interpret the permittivity increments (i.e., comparable to relative permittivity)?
TDR measurements are affected by the density and wetness of the medium they are placed in. Although the objective of using TDR probes is to detect liquid water flow within the snowpack, we are interested in comparing the evolution of the relative permittivity between two probes placed in the same layer. By dividing each measurement value by the first measurement of the time series, both curves start from the same point, making the comparison of their variations easier. Data normalization is a common practice in such circumstances. This is now better explained in the manuscript.

Figure 4 – I'm surprised that the permittivity values don't capture the seasonal densification of the snowpack. Any ideas for why this might be the case?
If we introduce the highest (0.35) and lowest (0.27) bulk relative density measured in the snow pits over the study period in Equation 6 in the manuscript ($\varepsilon_d = 1 + 1.7\rho_d + 0.7\rho_d^2$), we obtain respectively 1.7 and 1.5 as

relative permittivity. After normalization, we obtain 1 and 1.1 for the entire study period. Such variation can not be visualized in Figure 4.
Moreover, unlike LWC, density-driven relative permittivity evolutions are not reversible. In other words, an increase in relative permittivity cannot be followed by a decrease where density changes only occur. We use those two elements to identify changes in normalized relative permittivity due to changes in LWC in Figure 4. This has been added to the manuscript (Note that equation 6 is now introduced in the TDR method section).

Figure 5 – add subplot labels; y-axis "height" has a typo in all subplots
Done.

Figure 5 – swap date format to match other figures
Done.

Figure 5 – x-axis in multiple subplots has repeating numbers (6 7 7 8 and 7 8 8 9). Why is this?
Sorry, this is our mistake. Figure 5 has been corrected.

Figure 5 – at what spatial resolution are the bulk relative permittivities calculated? Add this to the Methods section.
Spatial resolution is a function of the drone speed and the sampling frequency. That information is now provided in the method section.

Figure 6 – Provide details on how the LWC results were kriged/interpolated? Consider overlaying the flight transects so the reader can better understand the data distribution.
The points used for the interpolation have been enlarged in the new version of Figure 6. In addition, the kriging procedure is now described in the document.

Figure 7 – This Figure has a lot of dead/white space, resulting in the actual results being relatively small. Please consider removing the y-axis tick labels for subplots b-d (i.e., after the first column) for each row and condensing each subplot to remove the white space to either side of the boxplots. The x-tick labels (flat, slope) could also likely be removed from all subplots other than the bottom row.
Done.

Figure 7 – swap date format to match other figures and add subplot labels
Done.

Figure 7 – Was the A2 WISe sensor used for LWC observations? If so, I would expect the dots to be colored black for snow pits. If not, what is the source for the LWC observations?
Yes, measurements were made with the A2 WISe sensor. Unfortunately, we mixed up the colour code for those measurements. This is now corrected.

Additional Figure: given the importance of the UAV radar, I strongly suggest adding in a two panel figure showing a radargram and the matching picked radargram.
We have introduced a new Figure showing radargram and matching picked radargram in the manuscript. This is indeed a helpful add-up.

This is an exciting early experiment on UAS GPR-measured snow depth which leads me to want to give some preference to this paper for its novelty. The goals in the abstract indicate there's a fairly robust snowpack monitoring setup along with acknowledging there are some inherent challenges to using GPR—which, yes, totally agree there are challenges. The article and data itself are more along the lines of "here's what we did" instead of "here's what we found." There are interesting ideas, but not sure the analysis did the dataset justice. For a highly quantitative topic, this paper is extremely qualitative.

The authors also mention several instances there is a degree of uncertainty beyond what the data directly presents (Line 165) which doesn't strike confidence with me either. Again, despite saying LWC in the abstract was well-measured, the authors reference in the results that it was highly overestimated (Line 356), so there is a lack of agreement and consistency that doesn't support this sense of confidence.

> We acknowledge that the way results were interpreted in the previous version of the manuscript was too vague and even contradictory at some points. In order to correct that weakness, we first clarified what the main findings of our research are:
> - The method we used allowed following the evolution of the bulk relative permittivity and the depth of the snowpack in time (between two surveys) and in space (in between the two surveyed areas).
> - Calculated *LWC,* relative density, and *SWE* absolute values are not matching those from reference methods in a satisfactory ways
> - The evolution of those three calculated variables in between surveys is realistic in dry conditions.
> - In wet conditions, the calculated *LWC* evolution only could be seen as representative of the natural variations. However, the lack of reference does not allow our study to be conclusive on that point.
>
> We then reviewed the entire document to ensure its content was aligned with those findings.

As someone who is most interested in the drone-based elements of this study, some more robust methods in comparing the GPR-measured snow depth against the DSM-measured snow depths are warranted. Even some sort of basic spatial correlation to identify where, if any, errors are present would be fascinating.

> As we use DSM-measured snow depths to calculate bulk relative permittivity, GPR-measured and DSM-measured snow depth are dependent variables. Therefore, we fear that comparing them in the manuscript could be misleading for the reader.

Simply summating, all errors were less than 3 cm is not a satisfactory result given that most of the shallow snowpack is in the 15-20cm range (10-25% error at 3cm).

> We acknowledge that the 3 cm error in shallow conditions is substantial. Therefore, we have added a paragraph in the discussion section to address that point and modified the rest of the manuscript accordingly.

Lastly, while the authors acknowledged that they observed different morphologies of snow, they seemingly treated all of the snow the same, despite directly stating there are differences in density and water content from those morphologies. It's good to say you need to use different equations, but it's another to do so (Line 375). It is also a weak finding that in your results and reflections you lack confidence in those same results. Again, having some sort of GPR weighting to compensate for different snow morphologies will produce different results.

> This is, here again, a valid point. There is definitively a need to re-asses the set of the equations we and many other studies used to estimate LWC, bulk density and SWE from relative permitivities. This could be done by differentiating between the snowpack conditions observed in snow pits or in AWS measurements. A new paragraph has been added to the manuscript on that aspect.

Below are numerous recommendations for edits, clarifications, and additions. Upon reflection, these cover much of the submission. I'd suggest that major analytical revisions are needed and a very different presentation of this work should result from clearer writing.

Line 10: Eliminate for instance, reverse ice and snow
> Done.

Line 13: Eliminate aims to
> Done.

Line 18: Replace with "weekly drone surveys", change sentence to active voice
> This has been rephrased in the new version of the manuscript.

Line 26: Awkward, rewrite; snow pack rather than snow cover? And throughout
> Done.

Lines 25-30: No mention of SWE? Seems like that would be logical for hydrologic purposes
> Hydrological properties of the snowpack are now presented in the first part of the introduction.

Line 30-35: Eliminate references to socioeconomic factors, focus is assessing snow-cover change, not its impacts
> This sub-section has been completely modified to focus on snow cover change observations.

Line 31. hydroelectricity production is not an "economic sector", do you mean hydroelectric power? Also missing a comma
> Done.

Line 36: Eliminate a in "anticipate a further"
> Done.

Line 36 to 42 – cite Cho et al. Cho, E., McCrary, R.R. and Jacobs, J.M., 2021. Future

Changes in Snowpack, Snowmelt, and Runoff Potential Extremes Over North America. *Geophysical Research Letters*, *48*(22), p.e2021GL094985.

 Done.

Lines 45-50: Introduce concepts earlier; statement is not a natural conclusion from the previous studies.

 The introduction section has been modified for that purpose.

Line 55 – Clarify what "With a centimeter scale accuracy," means – is this vertical or horizontal; there are also numerous studies besides the one cited.

 This has been clarified and Avanzi et al., 2018 is now cited.

Lines 56-57: I'd disagree with that assertion, different word-choice needed

 Done.

Line 57. Better citations are Harder et al. 2020 and Jacobs et al. 2021 Harder, P., Pomeroy, J.W. and Helgason, W.D., 2020. Improving sub-canopy snow depth mapping with unmanned aerial vehicles: lidar versus structure-from-motion techniques. *The Cryosphere*, *14*(6), pp.1919-1935.
Jacobs, J.M., Hunsaker, A.G., Sullivan, F.B., Palace, M., Burakowski, E.A., Herrick, C. and Cho, E., 2021. Snow depth mapping with unpiloted aerial system lidar observations: a case study in Durham, New Hampshire, United States. *The Cryosphere*, *15*(3), pp.1485-1500.

 Done.

Throughout: Replace h with Snow Depth

 This has been done and $\rho$ has been replaced with snow density too.

Line 68: Define moderate precision

 Done.

Line 79: If it has much attention, then where's the citation of studies? References needed to validate assertion

 Done.

Line 82: Awkward, rewrite

 Done.

Line 90-91: Develop this thought more. Recall why they don't work and the opportunity presented. Why the weird indent of just this sentence? What is a representative scale?

 The entire sub-section on *LWC* measurement has been modified according to this comment.

Throughout: Unless the abbreviation is referencing first letters, replace with the actual name of variable

 Done.

Line 92 Rework this paragraph through the rest of the introduction; GPR needs to be introduced earlier then the relationship between GPR and snow density. In order to motivate the study. Also forests are mentioned in several places but it isn't clear if the study seeks to capture forest snowpacks.

The introduction has been modified to introduce GPR snow studies earlier in the manuscript. In addition, a modification was made to define the study purpose earlier in the manuscript.

Line 93. permittivity is not mentioned previously or defined, there is no literature cited describing GPR and snow

Done. Literature describing GPR and snow has been added in Line 65 (Previati et al. 2011.)

Line 95: State why LWC can be neglected

By definition, a dry snowpack does not contain liquid water content. Therefore, *LWC* in dry conditions is equal or can be assumed to be 0%. This is now clarified in the manuscript.

Line 96: Why is snow density abbreviated, but relative permittivity not? Only one assumption is mentioned, why either?

Our mistake. This is now corrected.

Line 99: Showing potential how?

By providing a new method to measure spatial variability of snowpack's hydrological properties without disturbing the snowpack. This is now clarified.

Lines 100-105: Major undercutting of research efforts, perhaps eliminate or decrease severe language

This part of the introduction has been modified in the new version of the manuscript.

Line 111: MASL or define acronym

Done.

Line 112: Topographic?

Done.

Line 113 clearance?

Our mistake, we meant "clearing", this has been modified in the new version of the document.

Lines 115-120: Rewrite thermometer methods. Use positive depths for clarity.

Negative depths were used to avoid confusion between snow and ground temperature. This is now modified in the new version of the manuscript.

Line 116: Eliminate thanks to

Done.

Line 118 How was snowpack temperature measured?

Our mistake, "Snowpack temperature was measured with four thermometers maintained from 0 to 30 cm above the soil at 10 cm depth interval." has been added in the new version of the manuscript.

Line 124: Replace that with the
    Done.

Line 125: Is this study only about the ablation period? Aren't they all of interest?
Rewrite
    The meteorological conditions section has been rewritten.

Line 127: Snowpack is snow...which is by nature, cold. What does that mean? Define
cold or cold content as well as initial conditions of the snowpack when the study
began.
    We defined a cold snowpack as a pack which average temperature is below the
    melting point. This has been precised in the new version of the manuscript.

Line 129: What was it before? Language is too colloquial
    This has been clarified in the new version of the article.

Line 130: Eliminate Interestingly
    Done.

Throughout: More precise language is needed. Seems colloquially written with many
arbitrary adjectives
Throughout: Instead of writing about the weather, perhaps a time series of snowpack
temperature?
    Modifications of the meteorological section have been made, with a more
    data-oriented description of the snowpack conditions.

Line 139: Decreased from 30% from what to what?
    The AWS section has been modified to add more specifics and detailed
    descriptions of the ROS events and their impacts on the snowpack.

Line 140: Note date of last survey, not "last survey"
    Done.

Line 142: Remained negative in F or C? Replace with "remained below ##F/C"
    This has been corrected for the new version of the manuscript.

Line 145-149: What is this referring to? What does quasi-impermeable mean in
context? Rewrite or eliminate
    Figure 2b shows that, at the AWS, the soil is under zero degree Celcius at -20
    cm and positive at -30 cm. We realize that we moved too fast from the results
    to interpretation without explaining our point. Therefore, precisions are now
    provided in that section.

Throughout: Refocus study site section to reflect the study site, not the specific
climate patterns which quickly blend amongst one another. Perhaps is best replaced
with a figure of temperature
    As presented earlier, we have restructured this aspect of the paper entirely.

Line 154. Single points can't be used to quantify accuracy
    This have been rephrased in the new version of the manuscript.

Line 156: Something more formal than "spots"

Done.

Line 157 Where were the probes located? Height above ground?
The exact probe height evolves with time as the snowpack settles. This is now detailed in the TDR monitoring section.

Line 191: Shouldn't DJI go at the beginning? Rewrite for clarity
Done.

Lines 190-195: Why were these packages installed? What necessitated that? If they were used instead, why mention the DJI manufacturer specifications
Mavic 2 Pro drones are not equipped PPK at purchase. By adding the TopoDrone improvement kit, each picture get a better georeferencing. The camera itself remains unmodified. This is now explained in the new version of the manuscript.

Line 195: Is it necessary to mention Pix4D twice?
This was modified in the new version of the manuscript.

Line 201 Is this the h error or the DSM error? How was this error determined? If this is h, then this is a very low error; were there in situ observations to confirm? See the standards from previous SfM papers. What is the grid size of the DSM?
We acknowledge that this section was confusing. We mixed here the GNSS error with the error of the DSM. DSM error is estimated to be less than 10 cm. The horizontal resolution is 0.6 cm/pixel. This section has been modified.

Line 214 What is "the algorithm"?
An automatic graphic interpretation tool was used to identify layers in the radargrams. This has been added to the manuscript

Line 255 Describe the kernel algorithm rather than referencing the GIS software.
The interpolation parameters, number of points, maximal distances at which data points are used for calculation and method have been added to the new version of the manuscript.

Line 266: Draw out what that purpose was. Being explicit likely helps frame the significance of the result
This has been rephrased in the new version of the manuscript.

Lines 270-275: Visually accurate is not appropriate. If there are no in situ snow depth measurements then there is no means to estimate the snow depth accuracy using SfM. Use consistent, non-subjective adjectives to describe snow depth. My idea of extra might be different than your own, for example
The presence of permanent features that were observables on all DSM were used to evaluate the DSM quality. These references as well has the possible presence of non identified artefacts in the studied areas of the DSM provides visual insight on the data quality. This have been explained in the new version of the manuscript.

Throughout: Specific language needed. A greater reliance on numerically-sourced explanation would improve the clarity of the paper. Too often the authors resort to

arbitrary language to describe the results. Provided this is a GPR paper, there had to have been intense quantitative methods, otherwise, this paper is just a description of snowpack GPR imagery.

Most of the result sub-sections have been modified to add more numerical-based explanations.

Figure 1.a: Could you perhaps use a more hi-res map? Color would be useful

Figure 1 has been modified accordingly.

Figure 1.b: What do the numbers mean? What are the bounding boxes displaying? Legend and scale are needed. Is there imagery that isn't as dark as this?

Figure 1 has been modified accordingly.

Figure 1.c: there needs to be a way to link this Figure to 1b. Maybe add instruments to this Figure? Match the sensor names in table 1 with this Figure; what is at the top of Figure 1b? a building? A vehicle? Remove these from the figures. Flight lines are needed in this Figure.

Figure 1 has been modified. In the new version red dashed polygons have been drawn to help the reader to link with Figure 1c. In addition, all DSM based figures have been modified according to this request.

Figure 2 – correct spelling; what is relative density?

Relative density is the density divided by the water density. This is now detailed in the AWS method section in the new manuscript.

Figure 3: Use a different color scale. This signifies elevation to me, not snow depth. Red-Blue is better; indicate what red lines are

This color scale provides is the one that provided the best base for interpretation among all the ones we tested. Moreover, using a different color scale for snow depth in Figure 3 than in Figure 5 could be confusing. We propose to keep it as it is.

Figure 4 which line is which?

Done.

Figure 5 How were these measured? What is the origin? These lines should be on Figure 1

These transects were measured by superposing UAV-based GPR with the photogrammetry snow depth measurements. The transects have been added in the new version of Figure 1.

Figure 6: Use a graduated color scale, WAY too many choropleth options to make sense of the differences

As we would like to use the same color scale for all dates, using a graduated color scale may be confusing. We propose to keep it as it is.

---

## Author Response (AR2)

The authors would like first to thank the editor for the comments provided. There is no doubt that those comments improve the manuscript.

Line 72-73: "Remote sensing represents an attractive alternative for that purpose". Many of the techniques described in the preceding paragraphs are remote sensing. Be more specific here as to the methods you're referring to, e.g. "airborne remote sensing"

Done.

Line 114: remove "is" between studies and provides

Done.

Line 126: remove the word "tentatively"

Done.

Line 132: write out acronym in first use: "Bassin Versant Experimental (BVE) Ste-Marthe"

Done.

Line 141: Should this say "February 26 corresponded to the end of the accumulation period…" instead of "onset"?

Yes, this has been modified.

Line 155: For equation, remove the period or use 'x' or '*' for multiplication, i.e. SWE=h×ρ

Done.

Line 194: remove period and space between "probe" and "value"

Done.

Line 296: Looking at Figure 3, I think this sentence should say "by" instead of "to", as in "…increased both the SWE and relative snow density by 15 mm equivalent and 0.05 mm, respectively". Also, shouldn't relative density be unitless?

Yes and yes, these two point have been modified.

Line 306: Not sure what is meant by "of almost M.E.2". I think this sentence could be reworded to "A second mild episode (M.E. 2) started on March 8 and ended March 12, the day of the third survey"

Part of the former sentence wasn't removed: more than three days "of almost uninterrupted above 0°C air temperatures". As Mild episodes are now defined as "more than 24-hour-long periods with continuous above-zero air temperatures", this as been modify as suggested.

Line 388: remove "almost"

Done.

Line 400: I suggest using "non-zero" instead of "non-nulls"

Done.

Figure 1d: "Altitude" typically refers to flight height above ground. I think these plots are showing elevation profiles, in which case I suggest changing the labels to "elevation"

Figure 8: Somewhere in the caption or figure it should say what the source of the data is, i.e. 8a snow depth results from protogrammetry, 8b LWC from drone-based GPR, etc. I suggest adding letters to each chart (a, b, c, d) with a corresponding description in the caption which relates back to the text.

Done.

Line 269-270: Equation (8) and (9) were miss written the exponent of the second LWC as been added: $\varepsilon'_s = \left(0.1LWC + 0.8LWC^2\right)\varepsilon'_w + \varepsilon'_d$, (8)